# Transformers from an Optimization Perspective

**Yongyi Yang**[*]
University of Michigan
yongyi@umich.edu

**Zengfeng Huang**
Fudan University
huangzf@fudan.edu.cn

**David Wipf**
Amazon Web Services
davidwipf@gmail.com

## Abstract

Deep learning models such as the Transformer are often constructed by heuristics and experience. To provide a complementary foundation, in this work we study the following problem: Is it possible to find an energy function underlying the Transformer model, such that descent steps along this energy correspond with the Transformer forward pass? By finding such a function, we can view Transformers as the unfolding of an interpretable optimization process across iterations. This unfolding perspective has been frequently adopted in the past to elucidate more straightforward deep models such as MLPs and CNNs; however, it has thus far remained elusive obtaining a similar equivalence for more complex models with self-attention mechanisms like the Transformer. To this end, we first outline several major obstacles before providing companion techniques to at least partially address them, demonstrating for the first time a close association between energy function minimization and deep layers with self-attention. This interpretation contributes to our intuition and understanding of Transformers, while potentially laying the ground-work for new model designs.

## 1 Introduction

Although deep learning has achieved extensive practical success [21, 22], model architectures are often motivated by heuristics and at times remain difficult to understand. As such, significant effort has recently be devoted to various ways of analyzing and reinterpreting popular deep network structures. Of particular note for our purposes here are assorted attempts to create a one-to-one association between deep model layers and the unfolded iterations of some optimization process designed to minimize an interpretable energy function [10, 11, 47, 49, 35]. In this way the forward pass can be viewed as computing, at least approximately, a representation with minimal energy, where model parameters dictate the form of this energy and can be trained via the backward pass for a downstream goal of interest, e.g., classification or regression, etc.

This so-called *unfolded optimization* perspective can provide insights into the nature of the inductive biases introduced by different architectures, while potentially serving as a guide for bespoke design choices instantiated through properties of the underlying energy function involved. In fact, modifications of the latter can lead to predictable model behaviors, which have in the past been tailored to incorporate useful regularization factors [53], avoid subtle degenerate solutions [49], or design effective new models altogether [1, 10].

Despite these successes, the majority of prior work in this area has either addressed relatively simple network structures such as MLPs [47], or else restricted consideration to a single model component in isolation from a larger system [35]. Consequently, complex models such as the widely-used Transformer [43] have been mostly ignored, in large part because of the difficulty involved in simultaneously mapping both the self-attention and feed-forward network (FFN) modules to an integrated optimization process that minimizes a single, unified energy function.

---

[*]Work completed during an internship at the AWS Shanghai AI Lab.

36th Conference on Neural Information Processing Systems (NeurIPS 2022).

We take significant steps towards extending the unfolded optimization perspective to a full Transformer stack via the following contributions:

- After providing background details and existing examples of unfolded optimization, Section 2 formalizes the four key challenges to mapping Transformer layers to optimization steps descending a well-specified energy function. These include: (i) handling self-attention, (ii) integrating heterogeneous Transformer layer modules (i.e., self-attention and FFN), (iii) accounting for non-linear activations, and (iv) allowing for arbitrary/unstructured weight matrices.

- Later, Section 3 derives an energy function that closely reproduces self-attention per Challenge (i), Section 4 rigorously derives convergence results which demonstrate how to approximately handle Challenge (ii) via a novel form of alternating minimization, and Section 5 provides explicit details for solving Challenge (iii) via proximal methods; consideration of Challenge (iv) is deferred to the supplementary along with additional supporting analyses.

- The above contributions culminate in an energy function that is minimized by Transformer layers when certain technical conditions hold. To provide further motivation, Section 6 empirically demonstrates that indeed this energy is minimized with real-world data even when technical conditions are difficult to formally verify.

Collectively, these results provide a complementary foundation for understanding Transformer models, and suggest plausible avenues for future enhancements.

## 2 Brief Intro to Deep Architectures Formed from Unfolded Optimization

The basic idea of the unfolded optimization perspective is to create a one-to-one correspondence between the layers of a deep model and the iterations of some algorithmic process designed to minimize a parameterized, model-specific energy function. In this way, the forward pass can be viewed as computing an approximation to the minimum energy solution. Meanwhile, assuming each forward iteration is differentiable w.r.t. the energy function parameters, we can pass gradients of some meta objective function (e.g., classification, regression, etc.) through the approximate minimization steps for the backward pass, which forms an interpretable bilevel-optimization process [7, 10, 20]. Overall then, unfolded optimization produces minimal energy representations, whereby the specific structure of the energy function is trainable per the relevant downstream application domain.

### 2.1 Mathematical Formulation

The bilevel optimization process described above can be expressed more formally as:

$$Y^*(W, X) = \arg\min_Y E(Y; W, X), \tag{1}$$

$$W^* = \arg\min_W \frac{1}{S} \sum_{i=1}^{S} \ell_i(\psi[Y^*(W, X_i)]). \tag{2}$$

Here $S$ is the size of dataset, $\ell_i$ is the $i$-th loss function and $X_i \in \mathbb{R}^{n \times d}$ is the $i$-th input instance (e.g., input sentence to a language model), where $n$ is the number of tokens and $d$ is the token dimensionality. We refer to $Y \in \mathbb{R}^{n \times d}$ as the hidden representation of the model, where $Y^*$ is the (ideal) model output (e.g. set of word representations output by a Transformer encoder) that minimizes to lower-level energy function[2] $E : \mathbb{R}^{n \times d} \to \mathbb{R}$ whose form is dictated by parameters $W$ (e.g., these could also correspond with the trainable parameters in a Transformer model). And finally, $\ell$ is a meta-loss function (e.g., the loss function used to train a Transformer) while $\psi : \mathbb{R}^{n \times d} \to \mathbb{R}^{n \times d}$ is an arbitrary transformation that converts hidden representations to the model output, which can also potentially have its own learnable parameters. Assuming $\partial Y^*(W, X)/\partial W$ and $\partial \ell / \partial Y^*(W, X)$ are computable, where the former involves passing gradients through the iterative steps used to approximate $Y^*(W, X)$, then the combined bilevel system can be optimized w.r.t. $W$.

Note that hereinafter if the energy function does not include inter-token interactions, we view it as a function of vectors, and we use lower-case characters $\mathbf{y}$ and $\mathbf{x}$ to represent an arbitrary row of $Y$ and

---

[2]Note that although we frequently the term "energy", the concept is different from what has been called "energy-based learning" in the past, where the energy is defined on the joint input and output spaces [2, 23]; in our case energy is just a function for explaining the forward pass.

$X$ for simplicity. Also, when clear from context, we omit writing the parameters of a function, for example we may write $E(Y)$ to reference $E(Y; W, X)$.

## 2.2 Related Work and Limitations

In this section we present related work on unfolded optimization, and in particular, analyze some of the relevant limitations that serve as motivation for our efforts and underscore the challenges involved.

***Optimization-Induced Feed-Forward Networks*** A wide variety of prior work has investigated feed-forward structures from an unfolded optimization perspective [10, 12, 16, 18, 41, 47]. Here we take a closer look at [47] which is based on proximal methods. The basic energy function in [47] is

$$E(\mathbf{y}) = \mathbf{1}^\top \tilde{\sigma}^*(W^{-\top}\mathbf{y}) - \left\langle f(\boldsymbol{x}), W^{-\top}\mathbf{y} \right\rangle - \frac{1}{2}\|\mathbf{y}\|^2, \tag{3}$$

where $\mathbf{1}$ is a vector with all entries being 1, $\tilde{\sigma}^*$ is the convex conjugate [36][3] of $\tilde{\sigma} : a \mapsto \int_0^a \sigma(r)\mathrm{d}r$, $\sigma$ is an activation function and $f(\boldsymbol{x})$ is some transformation of input presentation $\boldsymbol{x}$. The proximal operator [8] of (3) is

$$\mathbf{y}^{(t+1)} = \mathrm{prox}_E\left(\mathbf{y}^{(t)}\right) = W^\top \sigma\left(W\mathbf{y}^{(t)} + f(\boldsymbol{x})\right), \tag{4}$$

which is analogous to the feed-forward layer of an MLP. Note that although $W$ can in principle be arbitrary matrix as long as the dimensionality is properly aligned, after stacking multiple layers of this model, the effective transformation is actually forced to be constrained:

$$\mathbf{y}^{(t+2)} = W^\top \sigma\left[WW^\top \sigma\left(W\mathbf{y}^{(t)} + f(\boldsymbol{x})\right) + f(\boldsymbol{x})\right], \tag{5}$$

where clearly after the first layer the actual feed-forward transformation becomes $WW^\top$, which is necessarily positive semi-definite (PSD). Hence unconstrained layer weights are not actually possible within this paradigm.

***Optimization-Induced Attention Mechanisms*** There is also limited work that attempts to interpret attention mechanisms from the unfolded optimization perspective [11, 35]. For example, the energy function

$$E(\mathbf{y}) = -\beta^{-1}\log\left(\sum_{i=1}^d e^{\beta S_i \mathbf{y}}\right) + \frac{1}{2}\|\mathbf{y}\|^2, \tag{6}$$

was proposed in [35], with subsequent updating using the concave convex procedure [51] leading to iterations of the form

$$\mathbf{y}^{(t+1)} = S\,\mathrm{softmax}(\beta S\mathbf{y}^{(t)}), \tag{7}$$

where $S \in \mathbb{R}^{n \times n}$ corresponds with the attention keys while $\mathbf{y}$ maps to the attention query, and $\beta$ is a constant scalar. Critically though, this attention mechanism is actually *cross-attention* (using $\mathbf{y}$ to attend to $S$), which does not well-align with the typical self-attention use-case of Transformers. Moreover, this work does not account for the aggregated Transformer feed-forward network module and attendant nonlinearity.

***Optimization-Induced Graph Neural Networks*** A variety of graph neural network architectures have also been developed from a similar optimization perspective [5, 26, 29, 32, 48, 49, 52, 54]. For example, graph attention mechanisms were derived using the iterative reweighted least squares (IRLS) algorithm in [49]. While this result is related to self-attention, which can be viewed as graph attention on fully connected graphs [4], it fails to produce the Transformer softmax term or the combined self-attention/feedforward Transformer stack. Nonetheless, we will later demonstrate how ideas from [49] can be leveraged to derive self-attention with softmax.

## 2.3 Key Challenges Extending to General Transformers

In this paper, we focus on a more complex model than prior work, namely, the Transformer encoder. A Transformer layer [43] is typically composed of two primary components: the self-attention layer and the feed-forward network (FFN) layer. If we simplify the FFN to a single linear transformation

---

[3]Here $\sigma^*$ is a scalar function, but for simplicity, we sometimes apply a scalar function to a vector or matrix in an entry-wise fashion for convenience.

with non-linear activation, and ignore the layer-norm and residual connections,[4] one Transformer layer reduces to the basic form

$$Y^{(t+1)} = \text{ReLU}\left[\text{softmax}\left(Y^{(t)}W_a Y^{(t)\top}\right) Y^{(t)}W_f\right], \tag{8}$$

where $W_a$ and $W_f$ represent self-attention and FFN layer weight matrices respectively. Regarding (8) and the limitations of prior work discussed above, we propose four major challenges in formally applying unfolded optimization to the Transformer setting:

1. *Token-Level Interactions*: As mentioned previously, Transformer models include not only a feed-forward process but also cross-token interactions, which is reflected by the self-attention mechanism considered to be the *sine qua non* of Transformers. And thus far, no prior work has derived Transformer-style self-attention, instead either relaxing to cross-attention [35], or failing to produce the ubiquitous softmax operator [49].

2. *Heterogeneous Layer Types*: Each Transformer encoder layer is composed of two totally different components: self-attention and a FFN. We refer to this structure as a "heterogeneous layer type," where each component has its own parameters and can be viewed as a distinct unfolded optimization process. However, it remains unknown whether or not it is possible to combine the corresponding energies of these two components to obtain a unified objective with any kind of convergence guarantees (even approximately) during the aggregated forward pass.

3. *Non-Linear Activations*: Many activation functions used in common neural network architectures can be understood from the perspective of proximal operators [24, 49], at least within the isolated context of some relatively simple feedforward schema. However, when integrated within the heterogeneous Transformer layer types mentioned above, the optimization process becomes rather complex, and it is unknown if any convergence properties still hold after the inclusion of a proximal step.

4. *Asymmetry of Weights*: As alluded to in the discussion of (5), the scope of most work addressing feed-forward networks is actually limited to models with symmetric (or even tighter, PSD) weight transformations [1, 10, 47], which restricts the resulting universality. Additionally, related models from the graph neural network literature are generally predicated on undirected graphs [48, 49], where the graph propagation is also symmetric. Consequently, although to our knowledge not addressed/discussed previously, it remains unknown how to construct energy functions for more general asymmetric transformations.

In this paper, we propose techniques to at least partially solve Challenges 1, 2 and 3, while for 4, we defer preliminary discussion to the supplementary, leaving formal investigation as a future direction.

## 3  A New Derivation of Transformer Self-Attention

We now tackle Challenge 1 by constructing an energy function whose iterative optimization steps match Transformer-style softmax self-attention on fully connected tokens as is customary.

### 3.1  Basic Softmax Self-Attention via Unfolded Optimization Steps

Consider the energy function

$$E_1(Y) = \sum_{i=1}^{n} \sum_{j=1}^{n} \rho\left(\frac{1}{2}\|\mathbf{y}_i - \mathbf{y}_j\|^2\right) + R(Y), \tag{9}$$

where $\mathbf{y}_i \in \mathbb{R}^{d \times 1}$ is the $i$-th row of matrix $Y \in \mathbb{R}^{n \times d}$, $\rho : \mathbb{R}^+ \to \mathbb{R}$ is a concave non-decreasing function, and $R : \mathbb{R}^{n \times d} \to \mathbb{R}$ is a convex function. Interestingly, although $E_1$ is not necessarily convex because of the non-convexity of $\rho$, under specific choices of $\rho$ and $R$, it can be optimized through a softmax-like structure as follows:

---

[4]Note that we are ignoring layer-norms and residual connections for simplicity of exposition; however, the analysis we present herein can naturally be generalized to accommodate layer-norm, and also certain forms of residual connections with additional assumptions; see supplementary for details.

**Theorem 3.1.** *Assume $\rho(z) = -\exp\{-z\}$, $R(Y) = \frac{1}{2}\|Y\|_{\mathcal{F}}^2$, and $\boldsymbol{\beta}_i = \exp\left\{-\frac{1}{2}\left\|\mathbf{y}_i^{(t)}\right\|^2\right\}$, and let $Y^{(t)}$ represent any fixed value for $Y$. Then the update rule*

$$\mathbf{y}_i^{(t+1)} = \frac{\sum_{j=1}^{n} \boldsymbol{\beta}_j \exp\left\{\mathbf{y}_i^{(t)\top}\mathbf{y}_j^{(t)}\right\} \mathbf{y}_j^{(t)}}{\sum_{j=1}^{n} \boldsymbol{\beta}_j \exp\left\{\mathbf{y}_i^{(t)\top}\mathbf{y}_j^{(j)}\right\}}, \quad \forall i, \tag{10}$$

*satisfies $E_1\left(Y^{(t+1)}\right) \leq E_1\left(Y^{(t)}\right)$ with equality iff $Y^{(t)}$ is a stationary point of $E_1$.*

It is worth mentioning that, although perhaps not obvious, the update step (10) can be generated via a majorization-minimization (MM) algorithm [42], where the majorization step produces a convex upper-bound, and the minimization step descends along the gradient of the upper-bound. Therefore, the core of this update is essentially a gradient step on a convex function, which will be relevant to the discussion in subsequent sections; see the proof and further details in the supplementary.

**Remark 3.2.** Several notable generalizations of Theorem 3.1 are possible. First, although in Theorem 3.1 we adopt a specific form for $\rho$ and $R$ to recover the softmax operator, with other selections of these functions the corresponding unfolded optimization algorithm can generate different/novel types of attention mechanisms. Secondly, obtaining (10) relies on a particular choice of gradient step size; however, for broader choices the resulting convergent updates interestingly lead to residual connections as a natural byproduct of the optimization trajectory (see supplementary). And finally, Theorem 3.1 can be easily modified to accommodate situations whereby full Transformer connectivity is constrained by some graph structure as in [6, 14] (again, see supplementary for details).

## 3.2 Extension to Include Trainable Parameters

After aggregating into matrix form, we have thus far shown that the iteration

$$Y^{(t+1)} = \text{softmax}_{\boldsymbol{\beta}}\left(Y^{(t)}Y^{(t)\top}\right)Y^{(t)}, \tag{11}$$

will reduce (or leave unchanged) the energy from (9), where $\text{softmax}_{\boldsymbol{\beta}}(\mathbf{y})_i = \frac{\boldsymbol{\beta}_i \exp\{\mathbf{y}_i\}}{\sum_j \boldsymbol{\beta}_j \exp\{\mathbf{y}_j\}}$ denotes a softmax operator with reweighting coefficient vector $\boldsymbol{\beta}$. If $\boldsymbol{\beta}_i$ is independent of $i$, i.e., $\left\|\mathbf{y}_i^{(t)}\right\|$ is constant (which is enforceable via layer normalization that can also be included within our framework as mentioned previously), then this reweighted softmax is equivalent to the canonical softmax used in Transformers.

Now consider the reparameterization $Y = ZW_a$, where $W_a \in \mathbb{R}^{d \times d}$ is an invertible matrix. It follows that $Z^{(t+1)}W_a = \text{softmax}_{\boldsymbol{\beta}}\left(Z^{(t)}W_a W_a^\top Z^{(t)\top}\right)Z^{(t)}W_a$, leading to the revised update

$$Z^{(t+1)} = \text{softmax}_{\boldsymbol{\beta}}\left(Z^{(t)}W_a^s Z^{(t)\top}\right)Z^{(t)}, \tag{12}$$

where $W_a^s = W_a W_a^\top$ and we have adopted the superscript 's' to indicate that this matrix is symmetric. Collectively then, the results of this section directly address Challenge 1, closely reproducing softmax-styled self-attention both with and without trainable parameters (the primary lingering limitation of symmetric weights being relegated to Challenge 4).

## 4 Combining Transformer Components via Alternating Inexact Minimization

We next address Challenge 2 and the heterogeneous Transformer layer types. With this goal in mind, we first introduce a general optimization scenario. Specifically, we ask the following question:

*Given two (convex) objectives $f(\mathbf{y})$ and $g(\mathbf{y})$, under what conditions, and/or to what extent, will alternatively taking separate gradient steps w.r.t. $f$ and $g$ optimize the aggregated function $f + g$?*

We refer to this optimization strategy as *alternating inexact minimization* (AIM), and we will shortly demonstrate conditions under which it converges to a ball with finite radius containing the optimal point of $f + g$. Later we discuss how these results contribute towards the resolution of Challenge 2.

We emphasize upfront that AIM as so-defined is quite different from what is commonly referred to as alternating minimization in the literature [15, 28]. The latter refers to scenarios whereby a unified objective function with multiple variables is minimized in an alternating fashion over one variable at a time with the others fixed, such that descent can be trivially achieved. In contrast, our AIM scenario involves multiple objective terms with a *shared* variable, and we alternate minimization over each term in isolation using the same variable, a much more challenging process to analyze.

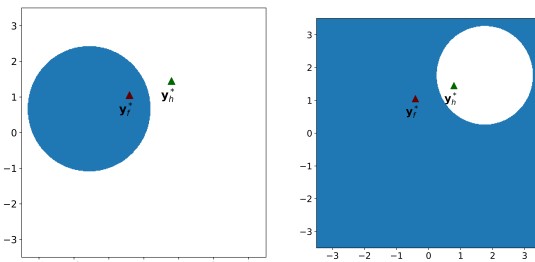 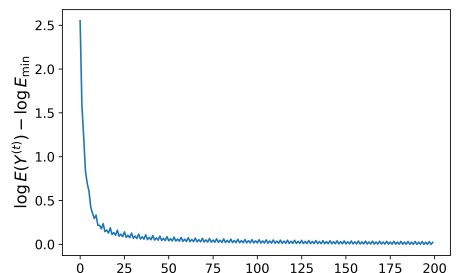

Figure 1: A two-dimensional illustration of $\mathcal{S}(\mathscr{C})$ with different values of $\mathscr{C}$. Left: $\frac{c_h \mathscr{C}}{L_f} = 0.7$. Right: $\frac{c_h \mathscr{C}}{L_f} = 1.5$. The blue area is $\mathcal{S}(\mathscr{C})$.

Figure 2: The value of energy $E\left(Y^{(t)}\right)$ with respect to iteration $t$. Note the y-axis is translated by a constant $\log E_{\min}$ to make the minimum align with zero.

## 4.1 General Alternating Inexact Minimization

Given two objectives $f, g : \mathbb{R}^d \rightarrow \mathbb{R}$, AIM is formalized as Algorithm 1, where $\alpha_1$ and $\alpha_2$ constitute step sizes. We will now investigate how Algorithm 1 relates to minimization of the combined objective defined by $h(\mathbf{y}) = f(\mathbf{y}) + g(\mathbf{y})$.

---

**Algorithm 1**

---

For the $t$-th iteration, execute $\quad \boldsymbol{u}^{(t)} = \mathbf{y}^{(t)} - \alpha_1 \nabla f(\mathbf{y}^{(t)}); \quad \mathbf{y}^{(t+1)} = \boldsymbol{u}^{(t)} - \alpha_2 \nabla g(\boldsymbol{u}^{(t)}).$

---

In the following, we assume that $f, g$ are both Lipschitz smooth and strongly convex [36], with Lipschitz constants $L_f$ and $L_g$ respectively, and convexity parameter $c_f$ and $c_g$ respectively. Therefore, $h$ will also be Lipschitz smooth and strongly convex, with Lipschitz constant $L_h$ and convexity $c_h$. We denote the optimal points of $f, g,$ and $h$ as $\mathbf{y}_f^*, \mathbf{y}_g^*,$ and $\mathbf{y}_h^*$ respectively.

Combining the two steps of Algorithm 1 gives

$$\mathbf{y}^{(t+1)} = \mathbf{y}^{(t)} - \alpha_1 \nabla f\left(\mathbf{y}^{(t)}\right) - \alpha_2 \nabla g\left[\mathbf{y}^{(t)} - \alpha_1 \nabla f\left(\mathbf{y}^{(t)}\right)\right], \tag{13}$$

while a canonical gradient descent step on objective $h$ is $\mathbf{y}^{(t+1)} = \mathbf{y}^{(t)} - \alpha_2 \left[\nabla f\left(\mathbf{y}^{(t)}\right) + \nabla g\left(\mathbf{y}^{(t)}\right)\right]$. Comparing the two update rules it is apparent that Algorithm 1 can be viewed as a noisy gradient descent step with step size $\alpha_2$ and noise factor

$$\Delta_t = \nabla h\left(\mathbf{y}^{(t)}\right) - \frac{\alpha_1}{\alpha_2} \nabla f\left(\mathbf{y}^{(t)}\right) - \nabla g\left[\mathbf{y}^{(t)} - \alpha_1 \nabla f\left(\mathbf{y}^{(t)}\right)\right]. \tag{14}$$

Noisy gradient descent is a well-studied problem in certain contexts [3, 31]. However, in our specific scenario we require a novel, different (actually tighter) bound than one can get from simply applying existing results.

Specifically, we demonstrate that when $\delta\left(\mathbf{y}^{(t)}\right) = \|\Delta_t\| / \left\|\nabla h\left(\mathbf{y}^{(t)}\right)\right\|$ is bounded, (13) is guaranteed to descend the objective $h$ as follows:

**Theorem 4.1.** *When $\alpha_1 \leq \alpha_2 \leq L_h^{-1}$, suppose $\mathbf{y}^{(t)}$ and $\mathbf{y}^{(t+1)}$ are related by (13), with $\delta\left(\mathbf{y}^{(t)}\right) \leq \mathscr{C}$ and $\mathscr{C} = \frac{\alpha_2}{\alpha_2 - \alpha_1 + \alpha_1 \alpha_2 L_g}$. Then $h\left(\mathbf{y}^{(t+1)}\right) \leq h\left(\mathbf{y}^{(t)}\right)$.*

**Remark 4.2.** Although the definition of $\mathscr{C}$ may appear rather complicated, when one of $\alpha_1, \alpha_2, L_f,$ or $L_g$ is sufficiently small, $\mathscr{C}$ behaves as $\Omega\left([\alpha_1 \alpha_2 L_f L_g]^{-1}\right)$.

We further consider how to interpret the constraint $\delta(\mathbf{y}) \leq \mathscr{C}$ and the region within which (13) can optimize $h$ as follows:

**Lemma 4.3.** *Let $\mathcal{S}(\mathscr{C}) = \left\{\mathbf{y} \,\middle|\, \frac{\|\mathbf{y} - \mathbf{y}_f^*\|}{\|\mathbf{y} - \mathbf{y}_h^*\|} \leq \frac{c_h \mathscr{C}}{L_f}\right\}$. When $\mathbf{y}^{(t)} \in \mathcal{S}(\mathscr{C})$, $\delta(\mathbf{y}) \leq \mathscr{C}$.*

Combining Theorem 4.1 and Lemma 4.3, we can therefore conclude that when $\mathbf{y}^{(t)} \in \mathcal{S}(\mathscr{C})$, $h\left(\mathbf{y}^{(t+1)}\right) \leq h\left(\mathbf{y}^{(t)}\right)$. Note that the boundary of $\mathcal{S}(\mathscr{C})$ given in Lemma 4.3 is called an Apollonian circle [37]. When $\mathscr{C} \leq \frac{L_f}{c_h}$, $\mathcal{S}(\mathscr{C})$ is a ball centered on $\mathbf{y}_f^*$, and when $\mathscr{C} \geq \frac{L_f}{c_h}$, $\mathcal{S}(\mathscr{C})$ is the whole space excluding a ball centered on $\mathbf{y}_h^*$. Figure 1 provides a 2D visualization for each case respectively.

Additionally, note that the same analysis can be done by switching the role of $f$ and $g$ (since the two processes are alternating) and further restrict the excepted region, although here we omit this for simplicity.

**Remark 4.4.** Our findings above can be summarized as follows: For sufficiently small values of $\alpha_1, \alpha_2$, Algorithm 1 reduces the combined objective $h$, at least provided that $\mathbf{y}$ is a certain distance away from the optimal point $\mathbf{y}_h^*$.

To illustrate this conclusion, we present a synthetic example with $f(Y) = \|SY\|_{\mathcal{F}}^2 + \|Y - B_1\|_{\mathcal{F}}^2$ and $g(Y) = \|YW\|_{\mathcal{F}}^2 + \|Y - B_2\|_{\mathcal{F}}^2$. Note that here we expand the variable $\mathbf{y}$ to a matrix $Y$ and let $f$ and $g$ consist of left and right transformations of $Y$ respectively, with an extra bias term to prevent the degenerate solution ($Y^* = 0$). We randomly set each entry of $\{S, W, B_1, B_2\}$ and execute Algorithm 1 with fixed step sizes and project the trace of $Y^{(t)}$ to a two-dimensional space with PCA for visualization. The trace of $Y^{(t)}$ is displayed in Figure 3 and the combined objective $h$ across iterations in Figure 2. From these plots, the behaviour predicted by our theory can be verified: When $Y^{(t)}$ is a sufficient distance away from $Y_h^*$ (the energy is relatively high), the optimization trajectory moves closer to $Y_h^*$ (descending the energy), and after $Y^{(t)}$ is close enough to $Y_h^*$ (and the $h$ is at a relatively low level), the energy oscillates in a certain range about the optimal solution.

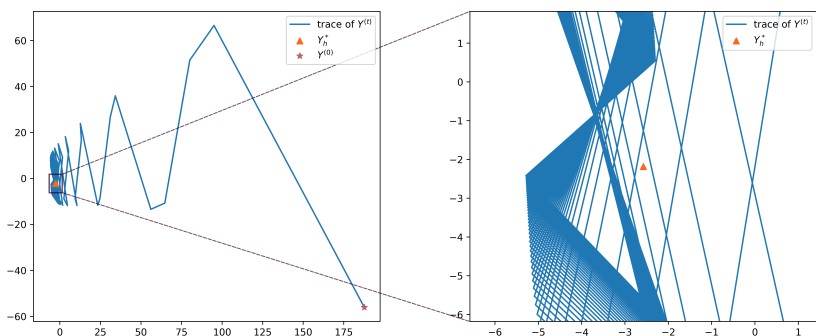

Figure 3: The trace of $Y^{(t)}$ projected to a plane. The figure on the right displays a zoomed local region around $Y_h^*$ taken from the figure on the left.

## 4.2 Unfolded Optimization of Heterogeneous Layer Type Models

We now return to Transformer-specific models. Let $E_2(Y) = \frac{1}{2}\mathrm{Tr}\left(YW_fY^\top\right) + \frac{1}{2}\|Y\|_{\mathcal{F}}^2$ and consider the combined energy $E(Y) = E_1(YW_a) + E_2(Y)$ (analogous to $h$ from the previous section) and alternatively perform the following AIM updates:

$$U^{(t)} = \mathrm{softmax}_{\boldsymbol{\beta}}\left(Y^{(t)\top}W_a^s Y\right)Y^{(t)}; \tag{15}$$

$$Y^{(t+1)} = U^{(t)} - \alpha_2\left.\frac{\partial E_2}{\partial Y}\right|_{Y=U^{(t)}} = U^{(t)}W_f^s, \tag{16}$$

where $W_f^s = (1 - \alpha_2)I - \alpha_2\frac{W_f + W_f^\top}{2}$ provides an additional linear transformation after the softmax.

From Section 3, we know that (15) is optimizing $E_1$ using essentially a row-wise gradient descent step after majorization. Likewise, (16) is also a gradient step with step size $\alpha_2$. Therefore, the combined rules fall into the scope of Algorithm 1, and from the analysis in Section 4.1, we can draw the following conclusion:

**Corollary 4.5.** *If $Y^{(t+1)}$ is computed via (15) and (16) with input $Y^{(t)}$, then $E\left(Y^{(t+1)}\right) \leq E\left(Y^{(t)}\right)$ when $Y^{(t)} \notin \mathcal{S}$, where $\mathcal{S}$ is a ball with finite radius containing $Y_{\hat{E}}^*$, the optimal point of $\hat{E}$, the convex upper bound of $E$.*

Note that after combining (15) and (16), the update aggregated update rule is already very similar to our target from (8), the only differences being the missing non-linearity and symmetric weights (note we use symmetric weight matrix $W_a^s$ and $W_f^s$ instead of $W_a$ and $W_f$ here), which correspond with Challenges 3 and 4 in Section 2.3 respectively; we address the former next.

# 5 Non-linear Activations Integrated within the Unfolding Paradigm

Handling non-linear activations within an unfolded optimization setting has been considered previously [12, 41, 45, 49]. However, prior work has largely relied on proximal operators to create non-linear activations paired with simple linear filters, and the analysis does not transfer to the more complex Transformer case under investigation here. While we will restrict our attention to adding ReLU activations in this section, both for simplicity of exposition and because of their ubiquity in Transformer models, our results generalize to broader selections.

Similar to Section 4, we first study a general form of the optimization problem by introducing a ReLU activations into Algorithm 1, formulating this modification as AIM with proximal steps, and then analyzing convergence under appropriate constraints. Later we apply these results to the Transformer.

## 5.1 Proximal-Alternating Inexact Minimization

---
**Algorithm 2**

---
For the $t$-th iteration, execute
$$\boldsymbol{u}^{(t)} = \boldsymbol{y}^{(t)} - \alpha_1 \nabla f\left(\boldsymbol{y}^{(t)}\right); \quad \boldsymbol{v}^{(t)} = \boldsymbol{u}^{(t)} - \alpha_2 \nabla g\left(\boldsymbol{u}^{(t)}\right); \quad \boldsymbol{y}^{(t+1)} = \mathrm{ReLU}\left(\boldsymbol{v}^{(t)}\right).$$

---

It has already been established [24, 41] that ReLU activations can be modeled as a proximal operator via

$$\mathrm{ReLU}(\mathbf{y}) = \arg\min_{\boldsymbol{z}} \frac{1}{2\lambda} \|\boldsymbol{z} - \mathbf{y}\|^2 + \phi(\boldsymbol{z}), \tag{17}$$

where $\phi$ is the indicator function for the first quadrant given by $\phi(z) = \begin{cases} +\infty & \text{if } z < 0 \\ 0 & \text{if } z \geq 0 \end{cases}$. Moreover,

in Section 4, we have demonstrated how the steps of Algorithm 1 together form an inexact gradient descent iteration of the loss $h(\mathbf{y}) = f(\mathbf{y}) + g(\mathbf{y})$ with noisy term $\Delta_t$ defined in (14). Here, with the addition of the proximal step in Algorithm 2, we obtain an inexact version of proximal-gradient descent. In fact, one turn of Algorithm 2 is equivalent to

$$\mathbf{y}^{(t+1)} = \arg\min_{\boldsymbol{z}} \frac{1}{2\lambda} \left\| \boldsymbol{z} - \mathbf{y}^{(t)} + \alpha_2 \nabla h\left(\mathbf{y}^{(t)}\right) - \alpha_2 \Delta_t \right\|^2 + \phi(\boldsymbol{z}), \tag{18}$$

which is an inexact version of canonical proximal-gradient step with noise $\Delta_t$, as compared to the exact version $\mathbf{y}^{(t+1)} = \arg\min_{\boldsymbol{z}} P\left(\boldsymbol{z}; \mathbf{y}^{(t)}\right)$, where $P(\boldsymbol{z}; \mathbf{y})$ is the proximal problem

$$P\left(\boldsymbol{z}; \mathbf{y}^{(t)}\right) = \frac{1}{2\lambda} \left\| \boldsymbol{z} - \mathbf{y}^{(t)} + \alpha_2 \nabla h\left(\mathbf{y}^{(t)}\right) \right\|^2 + \phi(\boldsymbol{z}). \tag{19}$$

While various forms of inexact proximal gradient descent have been studied in the past [9, 13, 38, 46], existing work either assumes constant noise [9], stochastic noise [46], or decreasing/convergent noise [13, 38]. Critically, no prior work that we are aware of applies to our case where the noise can potentially increase with iterations. Moreover, existing analysis in the literature is primarily concerned with convergence to a fixed point, while in our scenario, we instead consider entering a specific region formed around certain points.

In addition to bounding $\delta\left(\mathbf{y}^{(t)}\right)$ as in Section 4, we also need to bound the similarity between the current position $\mathbf{y}^{(t)}$ and the gradient $\alpha_2 \nabla h\left(\mathbf{y}^{(t)}\right)$, which is defined as $\mathfrak{D}(\boldsymbol{\xi}_1, \boldsymbol{\xi}_2) = \frac{1}{\|\boldsymbol{\xi}_1\|^2} \sum_{i=1}^{d} \min(\boldsymbol{\xi}_{2,i}^2 - \boldsymbol{\xi}_{1,i}^2, 0)$. Intuitively, $\mathfrak{D}(\boldsymbol{\xi}_1; \boldsymbol{\xi}_2)$ is defined such that each term of the summation is negative, but only close to 0 when $\boldsymbol{\xi}_{1,i}$ and $\boldsymbol{\xi}_{2,i}$ are both large. We then have the following:

**Theorem 5.1.** *If* $\alpha_1 \leq \alpha_2 \leq L_h^{-1}$, $\mathfrak{D}\left(\alpha_2 \nabla h\left(\mathbf{y}^{(t)}\right); \mathbf{y}^{(t)}\right) \geq -\kappa$ *for any* $\kappa \in (0, 1)$*, and* $\delta\left(\mathbf{y}^{(t)}\right) \leq \mathscr{C}'$*, where* $\mathscr{C}' = \frac{\alpha_2 c_P \lambda \sqrt{1-\kappa}}{\sqrt{2}(\alpha_2 - \alpha_1 + \alpha_1 \alpha_2 L_g)}$*, we have that* $h\left(\mathbf{y}^{(t+1)}\right) + \phi\left(\mathbf{y}^{(t+1)}\right) \leq h\left(\mathbf{y}^{(t)}\right) + \phi\left(\mathbf{y}^{(t)}\right)$.

Intuitively, Theorem 5.1 shows that the region where the descent of $h(\mathbf{y}) + \phi(\mathbf{y})$ is guaranteed is the intersection of $\mathcal{S}(\mathscr{C}')$ with $\mathcal{S}$ defined in Lemma 4.3, and the area $\mathcal{T}(\kappa) = \{\mathbf{y} \,|\, \mathfrak{D}(\alpha_2 \nabla h(\mathbf{y}); \mathbf{y}) \geq -\kappa\}$. While the shape of $\mathcal{T}(\kappa)$ remains difficult to specify in general, we note that $\mathcal{T}(\kappa)$ tends to the whole space when $\alpha_2 \to 0$ or $\kappa \to 1$. For example, we illustrate the area of $\mathcal{T}(\kappa)$ using a 2D synthetic example, where the energy function is $h(\mathbf{y}) = \|W\mathbf{y}\|_{\mathcal{F}}^2 + \|\mathbf{y} - \boldsymbol{b}\|^2$, and entries of $W$ and $\boldsymbol{b}$ are randomly generated. See Figure 4 for the visualization using different values $\kappa$, which shows that when $\kappa$ is sufficiently small, $\mathcal{T}(\kappa)$ is nearly the whole space (except a small area around the origin) which ensures the descent of $h(\mathbf{y}) + \phi(\mathbf{y})$ in most cases.

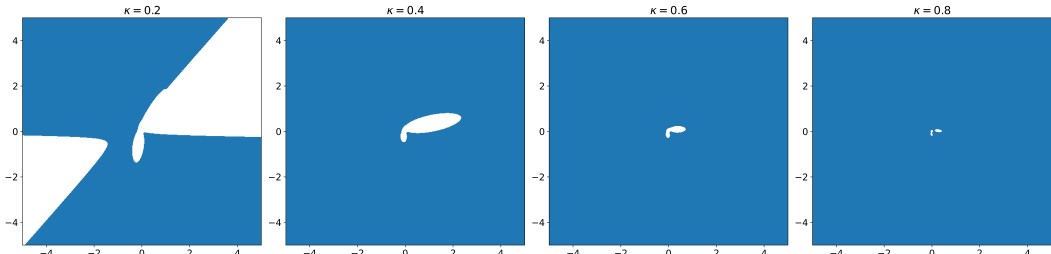

Figure 4: $\mathcal{T}(\kappa)$ region (blue area) of a 2D synthetic example with different values of $\kappa$.

## 5.2 Embedding the Non-Linearity within Transformer Models

By adding the penalty term to the previous energy, and with $E_1$ and $E_2$ obtained from Sections 3.1 and 4.2 respectively, we finally arrive at the total Transformer energy $E(Y) = E_1(Y) + E_2(Y) + \phi(Y)$. And the unfolded optimization of $E(Y)$ falls into the scope of Algorithm 2, such that the former analysis and Theorem 5.1 apply and we may conclude that the aggregated update rule

$$Y^{(t+1)} = \mathrm{ReLU}\left[\mathrm{softmax}_{\boldsymbol{\beta}}\left(Y^{(t)}W_a^s Y^{(t)\top}\right)W_f^s\right] \tag{20}$$

is a descent algorithm of $E$, except a region with finite measure. Moreover, if we modify the underlying gradient descent algorithm to make the step sizes $\alpha_1$ and $\alpha_2$ sufficiently small, the size of the exception region will tend to zero, and the update rule will be equipped with a residual term (see supplementary; similarly for a discussion about $\boldsymbol{\beta}$). Besides these issues, the unfolded update (20) only differs from (8) in its reliance on symmetric weights. This corresponds with Challenge 4 that we have not fully solved, although it has been shown that symmetric weights can mimic asymmetric weights if we enlarge the representation dimensions [19, 50].

## 6 Practical Verification

Although we have rigorously derived convergence criteria whereby Transformer layers descend a well-specified energy function to a region around optimal solutions, the analysis admittedly relies on conditions that would be difficult to formally verify on real-world datasets. However, our results are nonetheless amenable to targeted empirical corroboration, whereby we can check if the proposed energy does in fact descend during the Transformer forward pass on typical benchmarks.

To this end, we implement a Transformer model as described previously, up to known limitations like symmetric weights. We apply this model to two benchmarks, IMDB [30] and SST2 [40], which are both commonly-used sentiment classification datasets that rely on Glove-840b-300d [33] as the word embedding. Figures 5 and 6 display the energy of the output of each layer of a Transformer (as defined in (8)) averaged over 200 randomly chosen samples in the test set. Figure 5 uses randomly initialized weights while Figure 6 involves weights trained for 2000 steps with SGD and learning rate 0.01. Additionally, for the trained model we change the term $\|Y\|_{\mathcal{F}}^2$ in $E_1$ to $\|Y - X\|_{\mathcal{F}}^2$ in order to avoid degenerate representations (as can sometimes occur in trained Transformers [39]), noting that this modification is equally-well covered by our theory and leads to a commonly-used form of residual connection in the resulting Transformer architecture. See supplementary for details.

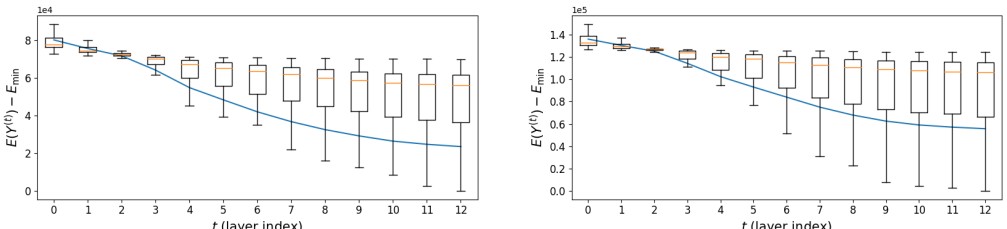

Figure 5: Box plot of the energy curves from randomly initialized 12-layer Transformers on IMDB (*left*) and SST2 (*right*) datasets; results are averaged over samples.

From these figures it is clear that even with real-world data, the the Transformer energy we have derived is (on average) monotonically decreasing across layers, matching the predictions of our

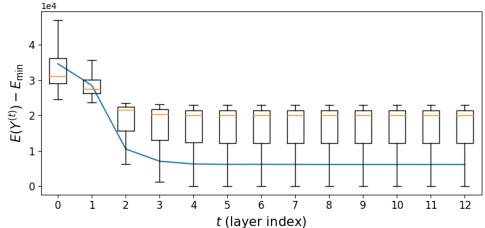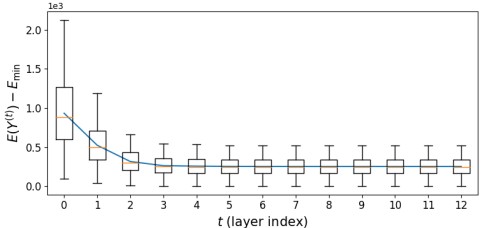

Figure 6: Box plots of the energy curves from trained 12-layer Transformers on IMDB (*left*) and SST2 (*right*) datasets; results are averaged over samples.

analysis. Moreover, with 12 layers, which represents a depth range not uncommon in practice (e.g., BERT), the model has not entered the fluctuation region. Moreover, this observation holds even with trained Transformer models composed from many/most of the typical components used in practice, namely, layers with self-attention, a linear transform followed by nonlinearity, and residual connections, etc. Hence although seemingly complicated, the conditions adopted by Lemma 4.3 and Theorem 5.1 are nonetheless likely hold in many practical settings.

## 7 Conclusion

While our contributions here have been primarily of a theoretical nature, there are nonetheless take-home messages that may be of practical relevance. First, as our overriding goal was to reproduce Transformer layers as closely as possible using an unfolded optimization perspective, very specific design choices were made in constructing the core underlying energy function. However, in practice we are free to choose alternative energies that can lead to different forms of tailored self-attention that may be be advantageous on an application-specific basis.

As a brief/simple representative example of the latter consider the following: The canonical Transformer uses softmax to normalize the attention coefficients. However, multiple works have questioned the appropriateness of softmax normalization, e.g., [27, 34]. Under our optimization unfolding perspective, we can see that the softmax normalization in the Transformer is derived from a specific choice of $\rho$ function, namely $\rho(z) = -e^{-z}$ (Section 3.1 and Theorem 3.1). But if we choose a different $\rho$, it would lead to a new normalization method other than softmax with interpretable properties. For instance, if $\rho(z) = \log(z + 2)$, then the attention coefficients would behave as (for simplicity we assume all $\mathbf{y}_i$'s have unit norm below):

$$a_{i,j} = \frac{1}{2 - \mathbf{y}_i^\top \mathbf{y}_j} \left( \sum_{k=1}^{n} \frac{1}{2 - \mathbf{y}_i^\top \mathbf{y}_k} \right)^{-1},$$

where $a_{i,j}$ is the attention coefficient between the $i$-th and $j$-th token. Since $\log(z + 2)$ grows slower than $-e^{-z}$ for $z \in [0, 1]$, the associated attention formulation would tend to encourage some more dissimilar representations between tokens. This is because in the new energy, a large value of $\|\mathbf{y}_i - \mathbf{y}_j\|$ (which means dissimilar $\mathbf{y}_i$ and $\mathbf{y}_j$) contributes less to the energy compared to before, and thus the optimization process has less motivation to reduce it. Conversely, if we choose a $\rho$ that grows faster than $-e^{-z}$ in the stated range (e.g. $\rho(z) = \log(z + 1)$), then the derived model would likely be encouraging more similar representations between tokens.

There are other ramifications of this overall framework with practical relevance as well. For example, the actual distribution of attention weights can potentially be better understood or influenced by properties of the energy function that produces them. Moreover, especially for regimes with limited data and therefore fewer free model parameters, the unfolding perspective can be used to devise architectures with inductive biases aligned with downstream tasks to help compensate for less model flexibility.

We conclude by noting that the techniques introduced in this paper can in some sense be viewed as universal tools for constructing a broader family of unfolding architectures. In fact, many/most deep learning models, including those with feed-forward structure [25], token-level interactions [44], residual connections [17], and heterogeneous layer types, can be reinterpreted through the lens of unfolded optimization using the framework we have introduced, at least up to some potential constraints like symmetric weights (which may be handled in other ways).

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
