# Supplementary File:
# Transformers from an Optimization Perspective

**Yongyi Yang**[*]
University of Michigan
yongyi@umich.edu

**Zengfeng Huang**
Fudan University
huangzf@fudan.edu.cn

**David Wipf**
Amazon Web Services
davidwipf@gmail.com

## A  Theory Background

This section provides background and references regarding some of the concepts and techniques used for proving our main results.

### A.1  Lipschitz Conditions and Strong Convexity

For a function $f : \mathbb{R}^d \to \mathbb{R}$, if

$$\exists L > 0, \forall \boldsymbol{x}, \mathbf{y} \in \mathbb{R}^d, \|f(\boldsymbol{x}) - f(\mathbf{y})\| \le L\|\boldsymbol{x} - \mathbf{y}\|, \tag{21}$$

we say $f$ satisfies the Lipschitz condition with Lipschitz constant $L$ or is simply $L$-Lipschitz. If *the gradient* of a function is $L$-Lipschitz, we say it is Lipschitz smooth with Lipschitz constant $L$. Moreover, if $f$ satisfies

$$\exists c > 0, \forall \boldsymbol{x}, \mathbf{y} \in \mathbb{R}^d, \|\nabla f(\boldsymbol{x}) - \nabla f(\mathbf{y})\| \ge c\|\boldsymbol{x} - \mathbf{y}\|, \tag{22}$$

we say it is (strongly) convex with convexity $c$, or simply $c$-strongly convex.

There are two important inequalities for Lipschitz smooth and strongly convex functions:

**Proposition A.1.** *If* $f : \mathbb{R}^d \to \mathbb{R}$ *is $L$-Lipschitz smooth, then*

$$\forall \boldsymbol{x}, \mathbf{y} \in \mathbb{R}^d, f(\mathbf{y}) \le f(\boldsymbol{x}) + \nabla f(\boldsymbol{x})^\top (\mathbf{y} - \boldsymbol{x}) + \frac{L}{2}\|\boldsymbol{x} - \mathbf{y}\|^2. \tag{23}$$

*If $f$ is $c$-strongly convex, then*

$$\forall \boldsymbol{x}, \mathbf{y} \in \mathbb{R}^d, f(\mathbf{y}) \ge f(\boldsymbol{x}) + \nabla f(\boldsymbol{x})^\top (\mathbf{y} - \boldsymbol{x}) + \frac{c}{2}\|\boldsymbol{x} - \mathbf{y}\|^2. \tag{24}$$

The proof of Proposition A.1, and more details of these two properties can be found in [2].

### A.2  Proximal Operators and Proximal Gradient Descent

The proximal problem of a function $\phi$ with parameter $\lambda$ is defined as

$$P(\boldsymbol{z}; \mathbf{y}) = \frac{1}{2\lambda}\|\boldsymbol{z} - \mathbf{y}\|^2 + \phi(\boldsymbol{z}), \tag{25}$$

and the mapping from $\mathbf{y}$ to the minimal point of $P(\cdot; \mathbf{y})$, is called the proximal operator of $\phi$:

$$\operatorname{prox}_\phi : \mathbf{y} \mapsto \arg\min_{\boldsymbol{z}} \frac{1}{2\lambda}\|\boldsymbol{z} - \mathbf{y}\|^2 + \phi(\boldsymbol{z}). \tag{26}$$

It can be shown that, for an objective function $h = f + \phi$, where $f$ is smooth, alternatively performing gradient descent on $f$ and proximal projection of $\phi$, i.e.

$$\mathbf{y}^{(t+1)} = \operatorname{prox}_\phi \left( \mathbf{y}^{(t)} - \alpha \nabla f\left( \mathbf{y}^{(t)} \right) \right), \tag{27}$$

is a descent algorithm of $h$, which is often referred to as proximal gradient descent [10].

---

[*]Work completed during an internship at the AWS Shanghai AI Lab.

36th Conference on Neural Information Processing Systems (NeurIPS 2022).

## A.3 Subgradient

For a (not necessarily smooth) convex function $f : \mathbb{R}^d \to \mathbb{R}$, it can be proven that the following set is always not empty [5]:

$$\partial f(\boldsymbol{x}) = \{\boldsymbol{g} | \forall \mathbf{y} \in \mathbb{R}^d, f(\mathbf{y}) \geq f(\boldsymbol{x}) + \boldsymbol{g}^\top (\mathbf{y} - \boldsymbol{x})\}. \tag{28}$$

We call $\partial f(\mathbf{y})$ the subdifferential of $f$ at point $\mathbf{y}$, and the elements of $\partial f(\mathbf{y})$ the subgradients of $\partial f$ at point $\mathbf{y}$.

Subdifferential satisfies linearity, which means

$$\alpha \partial f(\mathbf{y}) + \beta \partial g(\mathbf{y}) = \partial(\alpha f + \beta g)(\mathbf{y}), \tag{29}$$

where the summation of two sets is defined as the elementary summation

$$\partial f(\mathbf{y}) + \partial g(\mathbf{y}) = \{\boldsymbol{a} + \boldsymbol{b} | \boldsymbol{a} \in \partial f(\mathbf{y}), \boldsymbol{b} \in \partial \in g(\mathbf{y})\}. \tag{30}$$

Although subdifferentials are sets, based on the linearity, it is easier to use the equation symbol $=$ instead of $\in$ to denote subgradient, e.g. $\partial f(\mathbf{y}) = \boldsymbol{a}$ means $\boldsymbol{a} \in \partial f(\mathbf{y})$.

One very useful property of the subgradient is that it can indicate the global minimum of a convex function :

**Proposition A.2.** *If $f$ is convex, then*

$$0 \in \partial f(\mathbf{y}) \iff \mathbf{y} \in \arg\min_{\boldsymbol{z}} f(\boldsymbol{z}), \tag{31}$$

which is quite straight forward from the definition of subgradient. More detailed introduction and discussion of subgradients can be found in [3, 5].

# B  Proofs, Discussion, and Extensions of Section 3 Results

We now provide a complement of Section 3 of the main paper, including the proofs and discussion. For Theorem 3.1, we prove a more generalized version, taking into account of the structure of attention and the step size, as mentioned in Remark 3.2.

To achieve this, we extend the techniques used in [12] and show how to construct an energy function whose iterative optimization steps match Transformer-style self-attention on arbitrary graph structures – including fully connected graphs.

Note that this section provides a more general framework that assumes a certain graph structure exists in token-level interactions, which covers some attention sparsification works like [6, 7]. And when the graph is fully connected, we obtain a normal Transformer structure as analyzed in the main paper.

## B.1  General Unfolded Optimization Results for Creating Self-Attention on Graphs

Suppose $\mathcal{G} = (\mathcal{V}, \mathcal{E})$ is an undirected graph, whose adjacency matrix is $A \in \mathbb{R}^{n \times n}$, incidence-matrix is $B \in \mathbb{R}^{m \times n}$, degree matrix is $D = \text{diag}(A) \mathbf{1}$, and Laplacian matrix is $\mathcal{L} = D - A = B^\top B$, where $n = |\mathcal{V}|$ and $m = |\mathcal{E}|$. We use a tuple $(u, v)$ to represent an edge in the graph that connects node $u$ and $v$. Furthermore, let there be an index for each edge, suppose the $i$th edge connects $\text{fr}(i)$ and $\text{to}(i)$ and the index of edge that connecting $(u, v)$ is $\text{ind}(u, v)$. Then, the incidence matrix $B$ can be defined as

$$B_{i,j} = \begin{cases} 1 & \text{if } j = \text{fr}(i) \\ -1 & \text{if } j = \text{to}(i) \\ 0 & \text{others.} \end{cases} \tag{32}$$

Consider a generalized version of the energy function (9):

$$E_1(Y) = \sum_{u,v \in \mathcal{E}} \rho\left(\frac{1}{2} \|\mathbf{y}_u - \mathbf{y}_v\|^2\right) + R(Y), \tag{33}$$

where $\mathbf{y}_u \in \mathbb{R}^{d \times 1}$ is the $u$-th row of matrix $Y \in \mathbb{R}^{n \times d}$, $\rho : \mathbb{R}^+ \to \mathbb{R}$ is a concave non-decreasing function and $R : \mathbb{R}^{n \times d} \to \mathbb{R}$ is a convex function.

We refer to [12] that Algorithm 3 optimizes $E_1$:

**Algorithm 3**

Execute the following two assignment operations in arbitrary order:

1. $\gamma_{u,v}^{(t)} = \left. \frac{\partial \rho(z^2)}{\partial z^2} \right|_{z^2 = \frac{1}{2} \left\| \mathbf{y}_u^{(t)} - \mathbf{y}_j^{(t)} \right\|^2}$;

2. $Y^{(t+1)} = Y'^{(t)}$ such that $\tilde{E}_1 \left( Y'^{(t)}, \Gamma^{(t)} \right) \leq \tilde{E}_1 \left( Y^{(t+1)}, \Gamma^{(t)} \right)$,

where $\Gamma^{(t)} \in \mathbb{R}^{n \times n}, \Gamma_{i,j}^{(t)} = \gamma_{i,j}^{(t)}$, and $\tilde{E}_1 (Y, \Gamma) = \sum_{u,v \in \mathcal{E}} \frac{1}{2} \gamma_{u,v} \| \mathbf{y}_u - \mathbf{y}_v \|^2 + R(Y)$.

**Proposition B.1** (Lemma 3.2 in [12])**.** *Let $Y^{(t)}$ be the input of Algorithm 3 and $Y^{(t+1)}$ be the output, then $E_1 \left( Y^{(t+1)} \right) \leq E_1 \left( Y^{(t)} \right)$.*

In Algorithm 3, Step 1 converts the non-linear function $\rho$ to edge weights in graphs, which correspond to attention weight matrices. If in Step 2 we use gradient descent, then the update rule is

$$Y^{(t+1)} = \alpha(I - B^\top \Gamma B)Y^{(t)} + \left[ (1-\alpha)I - \frac{\partial R \left( Y^{(t)} \right)}{\partial Y^{(t)}} \right] Y^{(t)}, \qquad (34)$$

where the attention weights are injected into coefficients $1 - B^\top \Gamma B$. However, this naive framework does not include softmax used in Transformers. In the following subsections we show that with a special choice of $\rho$ and a reweighting of each row of $Y$, we get exactly self-attention updates with softmax.

Now, consider $\rho \left( z^2 \right) = - \exp \left\{ -z^2 \right\}, R(Y) = \frac{1}{2} \| Y \|_{\mathcal{F}}^2$. Then Step 1 gives

$$\begin{aligned} \gamma_{u,v}^{(t)} &= \exp \left\{ -\frac{1}{2} \left\| \mathbf{y}_u^{(t)} - \mathbf{y}_v^{(t)} \right\|^2 \right\} \\ &= \exp \left\{ \mathbf{y}_u^{(t)\top} \mathbf{y}_v^{(t)} \right\} \boldsymbol{\beta}_u \boldsymbol{\beta}_v, \end{aligned} \qquad (35)$$

where $\boldsymbol{\beta}_u = \exp \left\{ -\frac{1}{2} \left\| \mathbf{y}_u^{(t)} \right\|^2 \right\}$. Note that $\gamma_{u,u} = e^0 = 1$.

For Step 2, consider using gradient descent with Jacobi preconditioning, from which we arrive at the following Theorem:

**Theorem B.2.** *Consider updating $\tilde{E}_1$ using a gradient step with step size $\alpha$ and Jacobi preconditioner $\mathcal{D}^{-(t)}$:*

$$Y^{(t+1)} = Y^{(t)} - \alpha \mathcal{D}^{-(t)} \left. \frac{\partial \tilde{E}_1 \left( Y, \Gamma^{(t)} \right)}{\partial Y} \right|_{Y = Y^{(t)}}, \qquad (36)$$

*where*

$$\mathcal{D}^{(t)} = \left. \frac{\partial^2 \tilde{E}_1 \left( Y, \Gamma^{(t)} \right)}{\partial Y^2} \right|_{Y = Y^{(t)}}, \qquad (37)$$

*and $\alpha \leq 1$, it follows that $\tilde{E}_1 \left( Y^{(t+1)} \right) \leq \tilde{E}_1 \left( Y^{(t)} \right)$. And the update rule in (36) can be written as*

$$\mathbf{y}_u^{(t+1)} = (1-\alpha)\mathbf{y}_u^{(t)} + \alpha \frac{\sum\limits_{v \in \tilde{\mathcal{N}}(u)} \boldsymbol{\beta}_v \exp \left\{ \mathbf{y}_u^{(t)\top} \mathbf{y}_v^{(t)} \right\} \mathbf{y}_v^{(t)}}{\sum\limits_{v \in \tilde{\mathcal{N}}(u)} \boldsymbol{\beta}_v \exp \left\{ \mathbf{y}_u^{(t)\top} \mathbf{y}_v^{(t)} \right\}}, \quad \forall u. \qquad (38)$$

Notice that it requires $\alpha \leq L^{-(t)}$ to guarantee (36) is a descent step of $\tilde{E}_1$, where $L^{(t)}$ is the Lipschitz constant of the gradient of $\mathcal{D}^{-(t)} \tilde{E}_1 \left( Y, \Gamma^{(t)} \right)$ [2]. And by the definition of $\mathcal{D}^{(t)}$, $L^{(t)}$ is always 1. Therefore, $\alpha \leq 1$ always satisfies the condition.

## B.2 The Proof of Theorem 3.1

In Theorem B.2, if we take $\mathcal{G}$ to be a complete graph, then $\tilde{\mathcal{N}}(u) = \{1, 2, \cdots, n\}$. Besides, if $\alpha = 1$, (38) becomes

$$\mathbf{y}_u^{(t+1)} = \frac{\sum_{v=1}^n \exp\left\{\mathbf{y}_u^{(t)\top}\mathbf{y}_v^{(t)}\right\}\boldsymbol{\beta}_v\mathbf{y}_v^{(t)}}{\sum_{v=1}^n \exp\left\{\mathbf{y}_u^{(t)\top}\mathbf{y}_v^{(t)}\right\}\boldsymbol{\beta}_v}, \tag{39}$$

which guarantees the descent of $\tilde{E}_1(Y)$ by Theorem B.2. And further by Proposition B.1, Theorem 3.1 is proved.

## B.3 Discussion

From the analysis above, we have the following observations. Firstly, if we set $\alpha$ to other values, say $\alpha = \frac{1}{2}$, then the residual term in (10) is maintained, which corresponds with the common use of skip connections in the self-attention. Secondly, by assuming arbitrary graphs $\mathcal{G}$ (as opposed to merely the fully connected case described in the main paper), this framework can naturally handle attention mechanisms with special structure as in [6, 7].

# C Proofs of Section 4 Results

In this section, we prove the propositions in Section 4.

## C.1 The Proof of Theorem 4.1

Firstly, from the Lipschitz smooth assumption of the objective functions, we can bound the norm of the noise term:

$$\begin{aligned}
\|\Delta_t\| &= \left\|\nabla f\left(\mathbf{y}^{(t)}\right) + \nabla g\left(\mathbf{y}^{(t)}\right) - \frac{\alpha_1}{\alpha_2}\nabla f\left(\mathbf{y}^{(t)}\right) - \nabla g\left(\mathbf{y}^{(t)} - \alpha_1\nabla f\left(\mathbf{y}^{(t)}\right)\right)\right\| \\
&\leq \left\|\left(1 - \frac{\alpha_1}{\alpha_2}\right)\nabla f\left(\mathbf{y}^{(t)}\right)\right\| + \left\|\nabla g\left(\mathbf{y}^{(t)}\right) - \nabla g\left(\mathbf{y}^{(t)} - \alpha_1\nabla f\left(\mathbf{y}^{(t)}\right)\right)\right\| \\
&\leq \left\|\left(1 - \frac{\alpha_1}{\alpha_2}\right)\nabla f\left(\mathbf{y}^{(t)}\right)\right\| + \alpha_1 L_g\left\|\nabla f\left(\mathbf{y}^{(t)}\right)\right\| \\
&= \left(1 - \frac{\alpha_1}{\alpha_2} + \alpha_1 L_g\right)\left\|\nabla f\left(\mathbf{y}^{(t)}\right)\right\| \\
&\leq \left(1 - \frac{\alpha_1}{\alpha_2} + \alpha_1 L_g\right)\delta\left(\mathbf{y}^{(t)}\right)\left\|\nabla h\left(\mathbf{y}^{(t)}\right)\right\|.
\end{aligned} \tag{40}$$

In the following we write $\nabla h$ for short to refer to $\nabla h\left(\mathbf{y}^{(t)}\right)$; similarly $\delta$ for $\delta\left(\mathbf{y}^{(t)}\right)$ and $\Delta$ for $\Delta_t$. From (40) we have

$$\|\Delta\|^2 \leq \left(1 - \frac{\alpha_1}{\alpha_2} + \alpha_1 L_g\right)^2 \delta^2 \|\nabla h\|^2, \tag{41}$$

and from Cauchy-Schwarz

$$\nabla h^\top\Delta \geq -\|\nabla h\|\|\Delta\| \geq -\left(1 - \frac{\alpha_1}{\alpha_2} + \alpha_1 L_g\right)\delta\|\nabla h\|^2. \tag{42}$$

Therefore, by Lipschitz smoothness and convexity assumptions, and the inequalities (40), (41) and (42), and notice that $\alpha_2 L_h \leq 1$, we have

$$h\left(\mathbf{y}^{(t+1)}\right) - h\left(\mathbf{y}^{(t)}\right)$$

$$\leq -\alpha_2 \nabla h^\top (\nabla h + \Delta) + \frac{L_h}{2}\alpha_2^2 \|\nabla h + \Delta\|^2$$

$$= -\alpha_2 \|\nabla h\|^2 - \alpha_2 \nabla h^\top \Delta + \frac{L_h \alpha_2^2}{2}\left(\|\nabla h\|^2 + \|\Delta\|^2 + 2\nabla h^\top \Delta\right)$$

$$= \alpha_2 \left[\left(\frac{L_h \alpha_2}{2} - 1\right)\|\nabla h\|^2 + \frac{L_h \alpha_2}{2}\|\Delta\|^2 + (L_h \alpha_2 - 1)\nabla h^\top \Delta\right] \tag{43}$$

$$\leq \alpha_2 \|\nabla h\|^2 \left[\frac{1}{2}L_h \alpha_2 \left(1 - \frac{\alpha_1}{\alpha_2} + \alpha_1 L_g\right)^2 \delta^2\right.$$

$$\left. + (1 - L_h \alpha_2)\left(1 - \frac{\alpha_1}{\alpha_2} + \alpha_1 L_g\right)\delta + \left(\frac{L_h \alpha_2}{2} - 1\right)\right].$$

While seemingly complex, if we define $a = \frac{1}{2}L_h \alpha_2 \in \left[0, \frac{1}{2}\right]$ and $b = 1 - \frac{\alpha_1}{\alpha_2} + \alpha_1 L_g \geq 0$, (43) can be rewritten as

$$\frac{1}{\alpha_2 \|\nabla h\|^2}\left[h\left(\mathbf{y}^{(t+1)}\right) - h\left(\mathbf{y}^{(t)}\right)\right] \leq ab^2\delta^2 + (1 - 2a)b\delta + (a - 1). \tag{44}$$

Clearly, when $\delta \in \left[\frac{a-1}{ab}, \frac{1}{b}\right]$, we have

$$ab^2\delta^2 + (1 - 2a)b\delta + (a - 1) \leq 0, \tag{45}$$

and $\delta \geq 0 \geq \frac{a-1}{ab}$ by definition. Therefore, we conclude that $h\left(\mathbf{y}^{(t+1)}\right) - h\left(\mathbf{y}^{(t)}\right) \leq 0$ is guaranteed when

$$\delta \leq \frac{1}{b} = \frac{\alpha_2}{\alpha_2 - \alpha_2 + \alpha_1 \alpha_2 L_g}. \tag{46}$$

## C.2 The Proof of Lemma 4.3

We only need to notice that $\nabla f\left(\mathbf{y}_f^*\right) = \nabla h\left(\mathbf{y}_h^*\right) = 0$, then by Lipschitz smoothness and convexity of the objectives,

$$\|\nabla f(\mathbf{y})\| \leq L_f \left\|\mathbf{y} - \mathbf{y}_f^*\right\| \text{ and } \|\nabla h(\mathbf{y})\| \geq c_h \left\|\mathbf{y} - \mathbf{y}_h^*\right\|. \tag{47}$$

Therefore, when $\mathbf{y} \in \mathcal{S}(\mathscr{C})$,

$$\delta(\mathbf{y}) = \frac{\|\nabla f(\mathbf{y})\|}{\|\nabla h(\mathbf{y})\|} \leq \frac{L_f \|\mathbf{y} - \mathbf{y}_f^*\|}{c_h \|\mathbf{y} - \mathbf{y}_h^*\|} \leq \mathscr{C}. \tag{48}$$

# D Proofs of Section 5 Results

For simplicity, in the following we use $\mathbf{y}$ to denote $\mathbf{y}^{(t+1)}$ and $\boldsymbol{x}$ to denote $\mathbf{y}^{(t)}$ this section. We also use $\boldsymbol{x}^*$ to denote the optimal solution of the proximal problem (19):

$$\boldsymbol{x}^* = \arg\min_{\boldsymbol{z}} P(\boldsymbol{z}; \boldsymbol{x}) = \text{ReLU}(\boldsymbol{x}). \tag{49}$$

Recall that

$$\mathbf{y} = \arg\min_{\boldsymbol{z}} \frac{1}{2\lambda}\left\|\boldsymbol{z} - \boldsymbol{x} + \alpha_2 \nabla h(\boldsymbol{x}) - \alpha_2 \Delta_t\right\|^2 + \phi(\boldsymbol{z}). \tag{50}$$

First we shall prove the bound of the subgradient of proximal problem (19) at the point of $\mathbf{y}$:

**Lemma D.1.**

$$\partial P(\mathbf{y}; \boldsymbol{x}) = -\frac{\alpha_2}{\lambda}\Delta_t \tag{51}$$

*Proof.* Since **y** is the optimal point of (50), we have

$$\frac{\partial}{\partial \mathbf{y}} \left( \frac{1}{2\lambda} \left\| \mathbf{z} - \mathbf{x} + \alpha_2 \nabla h(\mathbf{x}) - \alpha_2 \Delta_t \right\|^2 + \phi(\mathbf{z}) \right) = 0, \tag{52}$$

which gives

$$\partial \phi(\mathbf{y}) = \frac{1}{\lambda} \left( \mathbf{x} - \mathbf{y} - \alpha_2 \nabla h(\mathbf{x}) + \alpha_2 \Delta(t) \right). \tag{53}$$

Therefore, the subgradient of (19) at **y** is

$$\partial P(\mathbf{y}; \mathbf{x}) = \frac{1}{\lambda} \left( \mathbf{y} - \mathbf{x} + \alpha_2 \nabla h(\mathbf{x}) \right) + \partial \phi(\mathbf{y})$$
$$= -\frac{\alpha_2}{\lambda} \Delta_t. \tag{54}$$

□

Then, we shall show the descent of $P(\mathbf{y}; \mathbf{x})$ to $P(\mathbf{x}; \mathbf{x})$ can be bounded by the distance between $\mathbf{x}$ and $\mathbf{x}^*$.

**Lemma D.2.**

$$P(\mathbf{x}; \mathbf{x}) - P(\mathbf{y}; \mathbf{x}) \geq \frac{c_P}{2} \left\| \mathbf{x} - \mathbf{x}^* \right\|^2 - \frac{\alpha_2^2 (1 + \alpha_1 L_g)^2 \delta(\mathbf{x})^2}{c_P \lambda^2} \|\nabla h(\mathbf{x})\|^2. \tag{55}$$

*Proof.* Let $c_P$ denote the convexity of proximal problem $P(\mathbf{z}; \mathbf{x})$. And note since $\mathbf{x}^*$ is the optimal point of $P(\mathbf{z}; \mathbf{x})$, we have $\partial P(\mathbf{x}^*) = 0$. Then by convexity we have

$$P(\mathbf{x}; \mathbf{x}) \geq P(\mathbf{x}^*) + \frac{c_P}{2} \left\| \mathbf{x} - \mathbf{x}^* \right\|^2 \tag{56}$$

and

$$P(\mathbf{x}^*) \geq P(\mathbf{y}; \mathbf{x}) + \partial P(\mathbf{y}; \mathbf{x})^\top (\mathbf{x}^* - \mathbf{y}) + \frac{c_P}{2} \left\| \mathbf{x}^* - \mathbf{y} \right\|^2$$
$$\geq P(\mathbf{y}; \mathbf{x}) - \|\partial P(\mathbf{y}; \mathbf{x})\| \, \|\mathbf{x}^* - \mathbf{y}\|$$
$$\geq P(\mathbf{y}; \mathbf{x}) - \frac{1}{c_P} \|\partial P(\mathbf{y}; \mathbf{x})\|^2$$
$$= P(\mathbf{y}; \mathbf{x}) - \frac{\alpha_2^2}{c_P \lambda^2} \|\Delta_t\|^2 \qquad \text{(By Lemma D.1)}$$
$$\geq P(\mathbf{y}; \mathbf{x}) - \frac{\alpha_2^2}{c_P \lambda^2} (1 + \alpha_1 L_g + \alpha_1 \alpha_2^{-1})^2 \delta(\mathbf{x})^2 \|\nabla h(\mathbf{x})\|^2. \qquad \text{(By (40))} \tag{57}$$

Combining (56) and (57) we prove the lemma.

□

Given Lemma D.2, we only need to bound $\|\mathbf{x} - \mathbf{x}^*\|$. The following lemma shows that, $\|\mathbf{x} - \mathbf{x}^*\|$ is related to

$$\mathfrak{D} : (\boldsymbol{\xi}_1, \boldsymbol{\xi}_2) \mapsto \sum_{i=1}^{d} \min \left( \boldsymbol{\xi}_{2,i}^2 - \boldsymbol{\xi}_{1,i}^2, 0 \right), \tag{58}$$

as stated in the main paper.

**Lemma D.3.** *Given* $\mathfrak{D} \left( \alpha_2 \nabla h(\mathbf{x}); \mathbf{x} \right) \geq -\kappa$, *we have*

$$\|\mathbf{x} - \mathbf{x}^*\|^2 \geq (1 - \kappa) \, \alpha_2^2 \|\nabla h(\mathbf{x})\|^2. \tag{59}$$

*Proof.* Note $\mathbf{x}_i^* = \sigma(\mathbf{x}_i - \alpha_2 \nabla h(\mathbf{x})_i) = \max(\mathbf{x}_i - \alpha_2 \nabla h(\mathbf{x})_i, 0)$. We have

$$\mathbf{x}_i^* - \mathbf{x}_i = \max(-\alpha_2 \nabla h(\mathbf{x})_i, -\mathbf{x}_i) = -\min(\alpha_2 \nabla h(\mathbf{x})_i, \mathbf{x}_i). \tag{60}$$

Therefore

$$
\begin{aligned}
\|\boldsymbol{x} - \boldsymbol{x}^*\|^2 &= \sum_{i=1}^{d} \min(\alpha_2 \nabla h(\boldsymbol{x})_i, \boldsymbol{x}_i)^2 \\
&\geq \sum_{i=1}^{d} \min(\alpha_2^2 \nabla h(\boldsymbol{x})_i^2, \boldsymbol{x}_i^2) \\
&= \sum_{i=1}^{d} \alpha_2^2 \nabla h(\boldsymbol{x})_i^2 + \sum_{i=1}^{d} \min\left(\boldsymbol{x}_i^2 - \alpha_2^2 \nabla h(\boldsymbol{x})_i^2, 0\right) \\
&= \alpha_2^2 \|\nabla h(\boldsymbol{x})\|^2 + \alpha_2^2 \|\nabla h(\boldsymbol{x})\|^2 \mathfrak{D}(\alpha_2 \nabla h(\boldsymbol{x}); \boldsymbol{x}) \\
&= \left(1 + \mathfrak{D}(\alpha_2 \nabla h(\boldsymbol{x}), \boldsymbol{x})\right) \alpha_2^2 \|\nabla h(\boldsymbol{x})\|^2 \\
&\geq \left(1 - \kappa\right) \alpha_2^2 \|\nabla h(\boldsymbol{x})\|^2
\end{aligned}
\tag{61}
$$

$\square$

Next, we shall show that $P(\boldsymbol{z}; \boldsymbol{x})$ is actually an upper bound of $f(\boldsymbol{z}) + \phi(\boldsymbol{z})$:

**Lemma D.4.** *If* $\lambda \leq \alpha_2 \leq \frac{1}{L_h}$, *there exists a function* $\eta(\boldsymbol{x})$, *which is only dependent on* $\boldsymbol{x}$, *which satisfies*

$$
P(\boldsymbol{z}; \boldsymbol{x}) + \eta(\boldsymbol{x}) \geq h(\boldsymbol{z}) + \phi(\boldsymbol{z}) \tag{62}
$$

*and*

$$
P(\boldsymbol{x}; \boldsymbol{x}) + \eta(\boldsymbol{x}) = h(\boldsymbol{x}) + \phi(\boldsymbol{x}). \tag{63}
$$

*Proof.* Let $\eta(\boldsymbol{x}) = h(\boldsymbol{x}) - \frac{\alpha_2^2}{2\lambda} \|\nabla h(\boldsymbol{x})\|^2$, then by the assumed Lipschitz condition we have

$$
\begin{aligned}
h(\boldsymbol{z}) + \phi(\boldsymbol{z}) &\leq h(\boldsymbol{x}) + \nabla h(\boldsymbol{x})^\top (\boldsymbol{z} - \boldsymbol{x}) + \frac{L}{2} \|\boldsymbol{z} - \boldsymbol{x}\|^2 + \phi(\boldsymbol{x}) \\
&\leq h(\boldsymbol{x}) + \nabla h(\boldsymbol{x})^\top (\boldsymbol{z} - \boldsymbol{x}) + \frac{1}{2\alpha_2} \|\boldsymbol{z} - \boldsymbol{x}\|^2 + \phi(\boldsymbol{z}) \\
&\leq \frac{1}{2\alpha_2} \left[ \|\boldsymbol{z} - \boldsymbol{x}\|^2 + 2\alpha_2 \nabla h(\boldsymbol{x})^\top (\boldsymbol{z} - \boldsymbol{x}) + \alpha_2^2 \|\nabla h(\boldsymbol{x})\|^2 \right] \\
&\quad + \phi(\boldsymbol{z}) - \frac{\alpha_2}{2} \|\nabla h(\boldsymbol{x})\|^2 + h(\boldsymbol{x}) \\
&\leq \frac{1}{2\lambda} \|\boldsymbol{z} - \boldsymbol{x} + \alpha_2 \nabla h(\boldsymbol{x})\|^2 + \phi(\boldsymbol{z}) - \frac{\lambda}{2} \|\nabla h(\boldsymbol{x})\|^2 + h(\boldsymbol{x}) \\
&= P(\boldsymbol{z}; \boldsymbol{x}) + \eta(\boldsymbol{x}).
\end{aligned}
\tag{64}
$$

And it is also straightforward to verify that

$$
P(\boldsymbol{x}; \boldsymbol{x}) + \beta(\boldsymbol{x}) = \frac{\alpha_2^2}{2\lambda} \|\nabla h(\boldsymbol{x})\|^2 + \phi(\boldsymbol{x}) - \frac{\alpha_2^2}{2\lambda} \|\nabla h(\boldsymbol{x})\|^2 + h(\boldsymbol{x}) = h(\boldsymbol{x}) + \phi(\boldsymbol{x}). \tag{65}
$$

$\square$

It is worth noting that, although there's a parameter $\lambda$ in the definition of $P$, when $\phi$ is defined as $\phi(z) = \begin{cases} +\infty & \text{if } z < 0 \\ 0 & \text{if } z \geq 0 \end{cases}$, which we used in the main paper to derive ReLU, the proximal operator $\text{prox}_\phi$ is independent of $\lambda$, which means we can always select a $\lambda$ small enough to satisfy $\lambda \leq \alpha_2$, which is required by Lemma D.4.

Lemma D.4 provides us with a majorization minimization perspective, which allow us to bound $h(\boldsymbol{z}) + \phi(\boldsymbol{z})$ by bounding $P(\mathbf{y}; \boldsymbol{x})$. And Lemma D.2 provides a way to bound the descent of $P(\mathbf{y}; \boldsymbol{x})$. Now we are ready to prove Theorem 5.1.

***The proof of Theorem 5.1*** Combining Lemmas D.2 and D.3 and recalling the bound of $\delta(\boldsymbol{x})$ we have

$$
P(\boldsymbol{x}; \boldsymbol{x}) - P(\mathbf{y}; \boldsymbol{x}) \geq \alpha_2^2 \|\nabla h(\boldsymbol{x})\|^2 \left[ \frac{c_P(1 - \kappa)}{2} - \frac{(1 + \alpha_1 L_g + \alpha_1 \alpha_2)^2 \delta(\boldsymbol{x})^2}{c_P \lambda^2} \right]. \tag{66}
$$

Given $\mathfrak{D}\left(\alpha_2 \nabla h\left(\mathbf{y}^{(t)}\right); \mathbf{y}^{(t)}\right) \geq -\kappa$ and (66), it follows that $P(\boldsymbol{x}; \boldsymbol{x}) - P(\mathbf{y}; \boldsymbol{x}) \geq 0$, i.e. $P(\boldsymbol{x}; \boldsymbol{x}) \geq P(\mathbf{y}; \boldsymbol{x})$.

By Lemma D.4,

$$h(\mathbf{y}) + \phi(\mathbf{y}) \leq P(\mathbf{y}; \boldsymbol{x}) + \eta(\boldsymbol{x}) \leq P(\boldsymbol{x}; \boldsymbol{x}) + \eta(\boldsymbol{x}) = h(\boldsymbol{x}) + \phi(\boldsymbol{x}), \tag{67}$$

which proves the theorem.

# E  Experiment Details

In the experiments of Section 6, the dataset and pre-processing scripts are provided by fastNLP[2]. The code to reproduce all experiments in the main paper is available[3] for the reference of the complete implementation of the experiments.

# F  Further Discussion

In this section, we provide further discussion related to symmetric weights, layer-dependent weights, layer normalization, and initial residual connections [4] as mentioned in the main paper.

## F.1  Asymmetric Weights

It initially seems hard or impossible to directly interpret an asymmetric transformation (i.e. $f(\mathbf{y}) = W\mathbf{y}$ where $W$ is asymmetric) as a gradient (See Lemma 5.3 in [13]). However, non-PSD weights are possible to be viewed as a gradient since

$$\frac{\partial \mathbf{y}^\top W \mathbf{y}}{\partial \mathbf{y}} = \left(W + W^\top\right) \mathbf{y}, \tag{68}$$

where $W + W^\top$ is symmetric but not necessarily PSD. Moreover, if the activation function is linear or meets certain criteria, it is also possible to use symmetric weights [13].

Furthermore, there is other work showing that models with symmetric weights are also universal approximators [8] or can have matching empirical results to models with asymmetric weights [9], which suggests that using symmetric weights might not adversely affect the expressivity of the model provided the hidden dimension can be increased sufficiently.

## F.2  Layer-dependent Weights

Note that since in the unfolding framework every step is the optimization process of a certain energy function, the model must exhibit a characteristic of recursive models like in (4), where the same weight matrix is shared at each layer. However, as has been discussed in various contexts, if the number of model layers is finite, it is possible to use shared weights to "simulate" layer-dependent weights [1, 11, 13]. Moreover, from the construction in [13], when the number of model layers is finite, the weights can even be asymmetric.

## F.3  LayerNorm

In the main paper, the model is actually a simplified version of the Transformer since LayerNorm is missing. Here we note that it is possible to combine LayerNorm in our proposed unfolding model. Apart from parameters, LayerNorm can be viewed as following process:

1. Translation: $\boldsymbol{u}^{(t)} = \mathbf{y}^{(t)} - \mathbf{1} \sum_{i=1}^{d} \mathbf{y}_i^{(t)}$,
2. Rescaling: $\mathbf{y}^{(t+1)} = \boldsymbol{u}^{(t)} / \left\|\boldsymbol{u}^{(t)}\right\|$;

which can both be interpreted as gradient steps:

---

[2]https://github.com/fastnlp/fastNLP
[3]https://github.com/FFTYYY/Transformers-From-Optimization

1. Translation: $\boldsymbol{u}^{(t)} = \mathbf{y}^{(t)} - \frac{\partial}{\partial \mathbf{y}} \left( \sum_{i=1}^{d} \mathbf{y}_i^{(t)} \right)^2$,

2. Rescaling: $\mathbf{y}^{(t+1)} = \boldsymbol{u}^{(t)} - \frac{\partial}{\partial \boldsymbol{u}^{(t)}} \left( \frac{1}{2} \left\| \boldsymbol{u}^{(t)} \right\|^2 - \left\| \boldsymbol{u}^{(t)} \right\| \right)$.

Therefore, using the techniques in Section 4, we can insert LayerNorm in an unfolding model by adding extra energy terms. Moreover, with LayerNorm applied, the reweighted softmax in (39) becomes normal softmax.

### F.4 Other forms of Residual Connections

Apart from the residual connections mentioned in Section B.3, it is also possible to derive other forms like initial residual connections that directly connect the input and current layer [4]. In order to achieve this, we can change the $R(Y)$ in (9) and Theorem 3.1 from $\frac{1}{2}\|Y\|_{\mathcal{F}}^2$ to $\frac{1}{2}\|Y - B\|_{\mathcal{F}}^2$, then the basic update in (11) becomes

$$Y^{(t+1)} = \text{softmax}_{\boldsymbol{\beta}} \left( Y^{(t)} Y^{(t)\top} \right) + B, \tag{69}$$

where $B \in \mathbb{R}^{n \times d}$ can be any bias term that is not depended on $Y$. For example if the entries of $B$ are learnable parameters, then it just changes the linear transformations to affine transformations, and if $B = f(X)$ where $f(X) \in \mathbb{R}^{n \times d}$ is some transformation of input features $X$, then $B$ serves as an initial residual connection just like the one used in [4]. The same can also be done with the $\|Y\|_{\mathcal{F}}^2$ term in $E_2$ defined in Section 4.2.