# OpenReview forum: "Transformers from an Optimization Perspective"
_NeurIPS.cc/2022/Conference — NeurIPS 2022 Accept_

### Official Review · Reviewer_siVJ · 2022-07-11

**Rating:** 5
**Confidence:** 3
**Soundness:** 3 good
**Presentation:** 3 good
**Contribution:** 2 fair

**Summary:**

To find an energy function underlying the Transformer model, the authors first outline 4 major obstacles and focus on solving or at least partly partially addressing them. The experimental results also help to support their results.

**Questions:**

How does the energy function underlying the Transformer help us to design or train the transformer? I’m interested in how useful the theorems are.

**Strengths And Weaknesses:**

Strength:
This paper is well-written. The background is easy to understand and friendly to the readers not in the unfolded optimization track. The problems the authors want to solve are also well-defined and explained clearly.
The novelty is also well claimed. The authors provided a clear comparison between this work and existing literature.
The experiments are also helpful to verify the viewpoints in this work.

Weakness:
The transformer architecture has been simplified. I worry about the impact of normalization layers and residual connection. Also, the order of different layers may also have an impact to the results. For instance, what would happen if the transformer used pre-LayerNorm or post-LayerNorm?

Similarly, what would happen if we use the widely used multi-head attention and other non-linear activation functions (e.g. GeLU)? The corresponding analysis is missing in this work.

In real-world applications, we usually use a pre-trained transformer instead of Glove+Transformer. Authors’ evaluation setting is a bit different from the real-world ones.

---

> ### Author Response · Authors · 2022-08-02
> **Response to Reviewer siVJ (Part 1)**
>
> Thanks for providing helpful comments/questions and recognizing the novelty and clarity of our work.  We address each question below.
>
> **The transformer architecture has been simplified. I worry about the impact of normalization layers and residual connection.  Also, the order of different layers may also have an impact to the results. For instance, what would happen if the transformer used pre-LayerNorm or post-LayerNorm?**
>
> While our analysis does not as of yet cover all of the many variants of Transformer models (and first-cut theoretical analyses rarely do), our core theory can be naturally extended to handle some forms of LayerNorm and residual connections.  For example, please see Sections C.3 and G.3 of the supplementary for a brief treatment of these extensions.
>
>
> **Similarly, what would happen if we use the widely used multi-head attention and other non-linear activation functions (e.g. GeLU)? The corresponding analysis is missing in this work.**
>
> Presently, we have only considered the case of single-head attention, and the extension to multi-head attention is non-trivial.  That being said, recent work suggests that multi-head attention is likely not the key to the success of Transformers, and single-head attention can actually achive comparable results [A,B,C]. Therefore, our single-head analysis thus far already captures perhaps the most important attributes of Transformers.  Still we can consider multi-head extensions for future work.  Good suggestion.
>
> As for other activation functions, although our submission focused on ReLU, the framework we present is also equally capable of incorporating any activation function that can be be expressed as a proximal operator of some penalty function. The latter requirement is not very restrictive though given that most commonly-used activations can be derived as proximal operators. (It has been shown in [D] that a function can be expressed as the proximal operator of some penalty term $\phi$ as long as it can be written as a subgradient of a convex function; for a continuous scalar function, this simply means it is non-decreasing, which is typical of Transformer activations.) Space permitting, we could include these ideas in an updated draft of the paper, or at least detail in the supplementary.
>
> As an illustration of the claim above for comparative purposes, here we provide the derivations for the commonly-used Leaky-ReLU activation: $\mathop{ \mathrm{ LeakyReLU } }(x) = \begin{cases} x & x \geq 0\\\\ a^{-1}x & x < 0\end{cases}$ where $a > 1$, can be written as a proximal operator in the following form: $$\mathop{ \mathrm{ LeakyReLU } }(x) = \arg\min_z \frac{1}{2}\\|z-x\\|^2 +\phi_{\text{lr}}(z),$$ where $\phi_{\text{lr}}(z) = \begin{cases}0 & z \geq 0 \\\\ \frac{1}{2}(a-1)z^2 &z < 0 \end{cases}$. Comparing this equation with (17) in the main paper, we can see that it differs with ReLU in the specific form of $\phi$.
>
> Note that while our framework has the universality to handle all element-wise, non-decreasing activation functions (as most activations used in practice are), the GeLU activation the reviewer mentioned represents a notable/rare exception, as this function can be decreasing over parts of its domain. We can consider generalizing our theory to handle decreasing activations in future work.
>
> [A] DeLighT: Deep and Light-weight Transformer, ICLR 2021 \
> [B] Multi-head or Single-head? An Empirical Comparison for Transformer Training, arxiv preprint \
> [C] Single Headed Attention RNN: Stop Thinking With Your Head, arxiv preprint \
> [D] A Characterization of Proximity Operators, Journal of Mathematical Imaging and Vision, 2020
>
> **In real-world applications, we usually use a pre-trained transformer instead of Glove+Transformer. Authors’ evaluation setting is a bit different from the real-world ones.**
>
> Given that ours is the first such rigorous analysis of Transformer unfolding, we cannot explicitly cover some real-world usage regimes.  In fact this is typical of almost all theoretical analysis, which invaribly relies on at least some simplifying assumptions.  Even so, in pre-training scenarios, we would nonetheless argue that our unfolding perspective still elucidates attributes of the pre-trained Transformer model itself, and could therefore potentially inform downstream applications; this could be an interesting direction for future work, thanks for the suggestion.  Note also that we have conducted new experiments with trained Transformers (see first section of revised Supplementary), and the results directly align with the predictions of our theory.

---

> > ### Author Response · Authors · 2022-08-02
> > **Response to Reviewer siVJ (Part 2)**
> >
> > **How does the energy function underlying the Transformer help us to design or train the transformer? I’m interested in how useful the theorems are.**
> >
> > This is a good question, similar to one asked by another reviewer, and helps to motivate why our framework is potentially very useful; we reproduce our response here.  And indeed it is possible to construct/design new network architectures/components inspired by the underlying Transformer unfolding perspective.
> >
> > As a brief/simple representative example, consider the following: The canonical Transformer uses softmax to normalize the attention coefficients. However, multiple works have questioned the appropriateness of softmax normalization, e.g., [A,B]. Under our optimization unfolding perspective, we can see that the softmax normalization in the Transformer is derived from a specific choice of $\rho$ function, namely $\rho (z) = -e^{-z}$ (Section 3.1 and Theorem 3.1). But if we choose a different $\rho$, it would lead to a new normalization method other than softmax with interpretable properties.
> >
> > For instance, if $\rho(z) = \log(z+2)$, then the attention coefficients would behave as (for simplicity we assume all $\mathbf y_i$'s have unit norm below):
> > $$a_{i,j} = \frac{1}{2-\mathbf y_i^\top \mathbf y_j}\left(\sum_{k=1}^n\frac{1}{2-\mathbf y_i^\top \mathbf y_k}\right)^{-1},$$where $a_{i,j}$ is the attention coefficient between the $i$-th and $j$-th token. And since $\log(z+2)$ grows slower than $-e^{-z}$ for $z \in [0,1]$, the associated attention formulation would tend to encourage some more dissimilar representations between tokens.  This is because in the new energy, a large value of $\\|\mathbf{y}_i-\mathbf{y}_j\\|$ (which means dissimilar $\mathbf{y}_i$ and $\mathbf{y}_j$) contributes less to the energy compared to before, and thus the optimization process has less motivation to reduce it. Conversely, if we choose a $\rho$ that grows faster than $-e^{-z}$ in the stated range (e.g. $\rho(z) = \log(z+1)$), then the derived model would likely be encouraging more similar representations between tokens.
> >
> >
> > Another example is, by changing the $R(Y)$ in (9) from $\\|Y\\|_\mathcal F^2$ to $\\|Y-X\\|_\mathcal F^2$ (as well as the $\\|Y\\|_\mathcal F^2$ term in $E_2$ of Section 4.2), we can derive an initial residual connection that directly connects input features with the current layer. This intial residual technique has been adopted in prior work (e.g. [C]) to avoid degenerate representations (known as over-smoothing in the field of GNNs), which can be a problem in Transformer models as well [D]. Such residual connections also emerge naturally from the unfolding Transformer framework we have presented.
> >
> > In any event these are just quick, illustrative examples, and future work can invariably explore numerous others motivated by the optimization unfolding framework. Note also that the general unfolding perspective is now ubiquitous across foundational models spanning machine learning, computer vision, and signal processing applications, with the conspicuous exception of Transformers because of the challenging integration of self-attention and FFN layers.  Hence our work can help to fill this notable gap even though we may not be able to precisely pinpoint what all of the future benefits may be.
> >
> > [A] SOFT: Softmax-free Transformer with Linear Complexity, NeurIPS 2021 \
> > [B] cosFormer: Rethinking Softmax In Attention, ICLR 2022 \
> > [C] Graph Neural Networks Inspired by Classical Iterative Algorithms, ICML 2021 \
> > [D] Revisiting Over-smoothing in BERT from the Perspective of Graph, ICLR 2022

---

### Official Review · Reviewer_9kiY · 2022-07-11

**Rating:** 6
**Confidence:** 5
**Soundness:** 3 good
**Presentation:** 3 good
**Contribution:** 3 good

**Summary:**

This paper proposes to understand Transformer through the lens of unfolding optimization. It overcomes several challenges in previous works of formulating the complex Transformer architecture and obtains several insights from the new interpretation.

**Questions:**

Please refer to the weakness part.

**Limitations:**

The author may want to elaborate on the implication of the proposed interpretation.

**Strengths And Weaknesses:**

Strengths:
- The formulation of the Transformer layer as unfolding optimization is novel in the Transformer literature and brings several new ideas, e.g., potential improvement of the architecture inspired by optimization techniques.
- The paper clearly states and tackles the challenge from previous works and is able to formulate the whole complicated Transformer layer.

Weakness:
- It would be great if the authors can further elaborate on how can we benefit from this new interpretation, e.g., any direct implication on architecture design.
- Is unclear if this analysis can be extended to a more practical setting of Transformer models, i.e., with components like LayerNorm.

---

> ### Author Response · Authors · 2022-08-02
> **Response to Reviewer 9kiY**
>
> Thanks for providing constructive comments/questions and pointing out the novelty of our work.  We address each question below.
>
> **It would be great if the authors can further elaborate on how can we benefit from this new interpretation, e.g., any direct implication on architecture design.**
>
> This is a good question, similar to one asked by another reviewer, and helps to motivate why our framework is potentially very useful; we reproduce our response here.  And indeed it is possible to construct/design new network architectures/components inspired by the underlying Transformer unfolding perspective.
>
> As a brief/simple representative example, consider the following: The canonical Transformer uses softmax to normalize the attention coefficients. However, multiple works have questioned the appropriateness of softmax normalization, e.g., [A,B]. Under our optimization unfolding perspective, we can see that the softmax normalization in the Transformer is derived from a specific choice of $\rho$ function, namely $\rho (z) = -e^{-z}$ (Section 3.1 and Theorem 3.1). But if we choose a different $\rho$, it would lead to a new normalization method other than softmax with interpretable properties.
>
> For instance, if $\rho(z) = \log(z+2)$, then the attention coefficients would behave as (for simplicity we assume all $\mathbf y_i$'s have unit norm below):
> $$a_{i,j} = \frac{1}{2-\mathbf y_i^\top \mathbf y_j}\left(\sum_{k=1}^n\frac{1}{2-\mathbf y_i^\top \mathbf y_k}\right)^{-1},$$where $a_{i,j}$ is the attention coefficient between the $i$-th and $j$-th token. And since $\log(z+2)$ grows slower than $-e^{-z}$ for $z \in [0,1]$, the associated attention formulation would tend to encourage some more dissimilar representations between tokens.  This is because in the new energy, a large value of $\\|\mathbf{y}_i-\mathbf{y}_j\\|$ (which means dissimilar $\mathbf{y}_i$ and $\mathbf{y}_j$) contributes less to the energy compared to before, and thus the optimization process has less motivation to reduce it. Conversely, if we choose a $\rho$ that grows faster than $-e^{-z}$ in the stated range (e.g. $\rho(z) = \log(z+1)$), then the derived model would likely be encouraging more similar representations between tokens.
>
>
> Another example is, by changing the $R(Y)$ in (9) from $\\|Y\\|_\mathcal F^2$ to $\\|Y-X\\|_\mathcal F^2$ (as well as the $\\|Y\\|_\mathcal F^2$ term in $E_2$ of Section 4.2), we can derive an initial residual connection that directly connects input features with the current layer. This intial residual technique has been adopted in prior work (e.g. [C]) to avoid degenerate representations (known as over-smoothing in the field of GNNs), which can be a problem in Transformer models as well [D]. Such residual connections also emerge naturally from the unfolding Transformer framework we have presented.
>
> In any event these are just quick, illustrative examples, and future work can invariably explore numerous others motivated by the optimization unfolding framework. Note also that the general unfolding perspective is now ubiquitous across foundational models spanning machine learning, computer vision, and signal processing applications, with the conspicuous exception of Transformers because of the challenging integration of self-attention and FFN layers.  Hence our work can help to fill this notable gap even though we may not be able to precisely pinpoint what all of the future benefits may be.
>
> [A] SOFT: Softmax-free Transformer with Linear Complexity, NeurIPS 2021 \
> [B] cosFormer: Rethinking Softmax In Attention, ICLR 2022 \
> [C] Graph Neural Networks Inspired by Classical Iterative Algorithms, ICML 2021 \
> [D] Revisiting Over-smoothing in BERT from the Perspective of Graph, ICLR 2022
>
>
> **Is unclear if this analysis can be extended to a more practical setting of Transformer models, i.e., with components like LayerNorm.**
>
> Our analysis can in fact be naturally extended to include practical components like LayerNorm (as well as residual connections).  Please see the supplementary (Sections C.3 and G.3) for a brief treatment of these topics.
>
> **The author may want to elaborate on the implication of the proposed interpretation.**
>
> Please see our previous responses above which address some of the implications of our work.

---

### Official Review · Reviewer_4boQ · 2022-07-11

**Rating:** 5
**Confidence:** 3
**Soundness:** 2 fair
**Presentation:** 3 good
**Contribution:** 2 fair

**Summary:**

The paper attempts to analyze the Transformer architecture from energy minimization perspective. In other words, authors seek an energy function (of input representations), such that minimizing this function is equivalent to running the forward pass of a corresponding transformer model. The paper lists the key challenges of building such energy function. Authors then address these challenges and propose an AIM-based procedure that corresponds to running the forward pass of a transformer with some additional constraints on model parameters.
Finally, the paper conducts two practical experiments where they show that transformers "as described previously, up to known limitations
like symmetric weights" actually reduce the corresponding energy function with subsequent model layers.



**Questions:**


### Minor corrections:
> (supplementary) ./code/README.md L5: python appllo_circle.py

perhaps, `apollo_circle` ? (typo)


Though authors have every right to decide for themselves, I would also recommend specifying the environment needed to run the supplementary code. Once could, for instance, add requirements.txt with specific library versions, or a docker container. This would simplify the task of reproducing the paper's results, especially in the far future.



**Limitations:**

The paper explicitly states the main limitations of their theoretical results (i.e. the requirement for symmetric weights).
However, I believe that the limitations of the proposed empirical results could be stated more clearly, so as not to mislead practically minded readers. I elaborate on these limitations in the "Strengths And Weaknesses" section.

**Strengths And Weaknesses:**

Authors propose a curious and novel (to my understanding) perspective on the popular transformer architecture.
To the best of my understanding, the overall approach looks sound, and the specific theoretical derivations for constructing the optimization procedure look solid. However, I am not an expert in the narrow domain of viewing and may have missed something.

My main concerns, however, are about practical applicability of the derived results.

### Section 6, Figures 5,6

Based on my understanding of the supplementary code (in particular, ./code/energy_curve/main.py , line 32-55) and the text presented in Section 6, __the energy figures 5,6 represent an untrained model.__

For instance, the figure with relu is constructed by the following code in ./code/energy_curve/main.py :

```
model_2 = Transformer(d , num_layers , 2)
model_2.normalize_weight()
paint(model_2 , "../generated_figures/%s.png" % relu_name)
```

The model is initialized, normalized, and immediately used for producing the plot. The training data is loaded a few lines above, but ultimately does not affect the plot in any manner except one: the model input dimension is based on word embedding (d). However, the embeddings themselves do not participate in updating model parameters.

Interestingly, __Section 6 does not explicitly contradict that__, but does not discuss it either. As such, the reader may be misled into believing that the practical experiments are more significant than they actually are.

To the best of my understanding, the actual evidence of practical applicability of this work is as follows:

1. all observations are made not with actual transformers, but with the proxy as defined in the paper (stated in Section 6)
    - in addition to the assumptions stated in the paper, there are other subtle differences, i.e. lack of scaling, or unconventional initialization
2. all observations are mad with randomly initialized weights, without any training on the data
3. it is unclear whether the said proxy retains the practical properties of actual transformers, i.e. whether it can actually solve the same tasks with the same quality
4. if it does, it is unclear whether the properties of the proxy transformer reflect the properties of actual transformer __once it has been trained on actual data__

With these four concerns in mind, it might be premature to claim that:

> Hence although seemingly complicated, the conditions adopted by Lemma 4.3 and Theorem 5.1 are nonetheless likely hold in many practical settings

> the unfolding perspective can be used to devise architectures with inductive biases aligned with downstream tasks to help compensate for less model flexibility

To summarize, I believe this paper proposes a curious theoretical perspective that may yet improve our understanding of real-world transformers. However, in its current for, it is unclear how well does the proposed approximation reflect the real world behavior of transformers, especially when trained.

---

> ### Author Response · Authors · 2022-08-02
> **Response to Reviewer 4boQ**
>
> Thanks to the reviewer for pointing out that our theoretical results are novel and sound.  In fact, the main criticism raised by the reviewer was limited to our verification experiments, not our primary theory per se.  In brief, the reviewer was concerned that our empirical results in Section 6 involved a randomly initialized Transformer instead of a trained model, and hence, this might reduce the practical relevance.
>
> To address this concern, we repeated the experiments from Section 6 with trained Transformer models, and the trends are the same as before.  More specifically, in each case we observe the induced energy function of the trained model descend across each layer, consistent with the predictions of our theory as expected.  While there is no space in the main paper for these new results and figures, we have added them to the beginning (Section A) of the updated supplementary file (perhaps later we can rearrange to fit in the main paper if the reviewer feels it is necessary).  Thanks for your comments, as resolving this issue certainly helps to further strengthen our paper and increase the practical relevance of our results.
>
>
> Other comments:
> + We appreciate the reviewer pointing out the typo in the readme file. It has been fixed in the revision.
> + In the "Limitations" section, the reviewer reiterated the potential limitation that our empirical results involved an untrained model.  However, given that we have reproduced the same phenomena using trained models (see above and Section A of the revised supplementary), we believe that this concern has been ameliorated.

---

> > ### Author Response · Authors · 2022-08-08
> > **Update for Reviewer 4boQ**
> >
> > We just wanted to double-check whether the updated experiments we provided (which are described in the rebuttal above and further detailed in the updated supplementary) are adequate for addressing the reviewer's initial concerns.  Note that the discussion phase will end soon, and after that we will no longer be able to answer follow-up questions or present additional experiments.  But we are happy to do so now if requested.

---

> > ### Comment · Reviewer_4boQ · 2022-08-09
> > **Rebuttal Acknowledgement**
> >
> > I have read the authors response and the updated supplementary materials and updated my score.
> > Authors have addressed one of my two main concerns about using untrained model.
> > I also agree that it is no longer necessary to update the limitations.

---

> > > ### Author Response · Authors · 2022-08-09
> > > **Response to Rebuttal Acknowledgement**
> > >
> > > Thanks for checking through our added experiments and providing updated feedback.  We also highlight that our primary contribution is theoretical, and in such cases there is inevitably some gap between theory and practical/deployable models. (As a quick representative example, there are numerous NeurIPS/ICML/ICLR papers analyzing the loss surface of deep *linear* neural networks, an architecture that has no appreciable practical value and yet still forms an influential starting point that contributes to our basic understanding of deep models.)
> > >
> > > Even so, although secondary to our main contribution, we believe that our experiments (including the new ones suggested by the reviewer which we have now added to the supplementary but will later move the the main paper space permitting) nonetheless provide valuable empirical support that significantly narrows the aforementioned gap.  In this regard, we point out that the trained Transformer models we used for the updated experiments had many/most of the typical components used in practice, namely, 12-layer models (as is BERT) comprised of self-attention, linear transform followed by nonlinearity, residual connections, etc.
> > >
> > > Anyway, thanks again for the reviewer's critique; certainly it has helped to improve our paper with stronger empirical support.

---

### Official Review · Reviewer_zDqk · 2022-07-12

**Rating:** 7
**Confidence:** 3
**Soundness:** 4 excellent
**Presentation:** 3 good
**Contribution:** 3 good

**Summary:**

This paper tried to find an energy function underlying the transformer model and reinterpret transformers as the unfoling of an interpretable optimization process across iterations. This paper starts from the complete Transformer architecture, and due to the structural complexity of the transformer, this paper addresses several challenges in the derivation process including heterogeneous layer types and non-linear activations.

**Questions:**

Please refer to the weaknesses.

**Limitations:**

No but not need.

**Strengths And Weaknesses:**

Strengths
- Detailed theoretical proofs are provided.
- Writing is logical and well laid out.

Weaknesses
- The approach of this paper seems to construct an energy function behind the network optimization process by analyzing the existing standard Transformer structure. Although the article mentions the practical value, this approach seems to be difficult to inspire the network design and the experimental part fails to provide the corresponding proof. Is there any actual demonstration of constructing a network out of the energy function in reverse?
- This paper mentions that the unfolding perspective has been adopted to analyze MLPs and CNNs. In terms of comparison at the network level, is there an opportunity to compare the energy function behind the different networks and the difference between their strengths and weaknesses?

---

> ### Author Response · Authors · 2022-08-02
> **Response to Reviewer zDqk (Part 1)**
>
> Thanks for providing insightful comments; we address each below.
>
> **The approach of this paper seems to construct an energy function behind the network optimization process by analyzing the existing standard Transformer structure. Although the article mentions the practical value, this approach seems to be difficult to inspire the network design and the experimental part fails to provide the corresponding proof. Is there any actual demonstration of constructing a network out of the energy function in reverse?**
>
> This is a good question and helps to motivate why our framework is potentially very useful. And indeed it is possible to construct new network architectures/components inspired by the underlying Transformer unfolding perspective.
>
> As a brief/simple representative example, consider the following: The canonical Transformer uses softmax to normalize the attention coefficients. However, multiple works have questioned the appropriateness of softmax normalization, e.g., [A,B]. Under our optimization unfolding perspective, we can see that the softmax normalization in the Transformer is derived from a specific choice of $\rho$ function, namely $\rho (z) = -e^{-z}$ (Section 3.1 and Theorem 3.1). But if we choose a different $\rho$, it would lead to a new normalization method other than softmax with interpretable properties.
>
> For instance, if $\rho(z) = \log(z+2)$, then the attention coefficients would behave as (for simplicity we assume all $\mathbf y_i$'s have unit norm below):
> $$a_{i,j} = \frac{1}{2-\mathbf y_i^\top \mathbf y_j}\left(\sum_{k=1}^n\frac{1}{2-\mathbf y_i^\top \mathbf y_k}\right)^{-1},$$where $a_{i,j}$ is the attention coefficient between the $i$-th and $j$-th token. And since $\log(z+2)$ grows slower than $-e^{-z}$ for $z \in [0,1]$, the associated attention formulation would tend to encourage some more dissimilar representations between tokens.  This is because in the new energy, a large value of $\\|\mathbf{y}_i-\mathbf{y}_j\\|$ (which means dissimilar $\mathbf{y}_i$ and $\mathbf{y}_j$) contributes less to the energy compared to before, and thus the optimization process has less motivation to reduce it. Conversely, if we choose a $\rho$ that grows faster than $-e^{-z}$ in the stated range (e.g. $\rho(z) = \log(z+1)$), then the derived model would likely be encouraging more similar representations between tokens.
>
>
> Another example is, by changing the $R(Y)$ in (9) from $\\|Y\\|_\mathcal F^2$ to $\\|Y-X\\|_\mathcal F^2$ (as well as the $\\|Y\\|_\mathcal F^2$ term in $E_2$ of Section 4.2), we can derive an initial residual connection that directly connects input features with the current layer. This intial residual technique has been adopted in prior work (e.g. [C]) to avoid degenerate representations (known as over-smoothing in the field of GNNs), which can be a problem in Transformer models as well [D].  Such residual connections also emerge naturally from the unfolding Transformer framework we have presented.
>
> In any event these are just quick, illustrative examples, and future work can invariably explore numerous others motivated by the optimization unfolding framework. Note also that the general unfolding perspective is now ubiquitous across foundational models spanning machine learning, computer vision, and signal processing applications, with the conspicuous exception of Transformers because of the challenging integration of self-attention and FFN layers.  Hence our work can help to fill this notable gap even though we may not be able to precisely pinpoint what all of the future benefits may be.
>
>  [A] SOFT: Softmax-free Transformer with Linear Complexity, NeurIPS 2021 \
>  [B] cosFormer: Rethinking Softmax In Attention, ICLR 2022 \
>  [C] Graph Neural Networks Inspired by Classical Iterative Algorithms, ICML 2021 \
>  [D] Revisiting Over-smoothing in BERT from the Perspective of Graph, ICLR 2022

---

> > ### Author Response · Authors · 2022-08-02
> > **Response to Reviewer zDqk (Part 2)**
> >
> > **This paper mentions that the unfolding perspective has been adopted to analyze MLPs and CNNs. In terms of comparison at the network level, is there an opportunity to compare the energy function behind the different networks and the difference between their strengths and weaknesses?**
> >
> > The unfolding of energy function $E_2$ (in line 242 of the main paper) gives a linear transformation:$$Y^{(t+1)} = Y^{(t)}-\alpha_2 \nabla E_2\left(Y^{(t)}\right) = W_f^s Y^{(t)},$$ which is the basic building block of MLPs (note that a bias term can also be introduced by replacing $\\|Y\\|_\mathcal F^2$ with $\\|Y-B\\|_\mathcal F^2$, which gives $Y^{(t+1)} = W_f^sY^{(t)}+B$). And the fact that $E_2$ forms a part of the Transformer energy is expected given that an MLP/FFN layer is a component of Transformer models.
> >
> > As for CNNs, if we define a CNN layer (ignoring activations and bias terms for simplicity) as  $$Y^{(t+1)} = CYW_f^s,$$ where $C$ is a symmetric convolution operator, then the corresponding energy function can be expressed as $$E_{\text{CNN}}(Y) = \mathrm{Tr} \left(YW_fY^\top C\right) + \frac{1}{2}\\|Y\\|_\mathcal F^2.$$  Comparing $E_\text{CNN}$ to $E_2$, it is clear that the major difference between them is that the former has an extra multiplication factor $C$. This extra factor provides the CNN with the ability to model inter-token interactions which the MLP itself doesn't have, but is compensated for by the attention terms in the Transformer noted in our paper.

---

> > > ### Comment · Reviewer_zDqk · 2022-08-09
> > > **Reply to authors feedback**
> > >
> > > Thanks for authors detailed feedback.
> > > That sounds reasonable to construct different attentional behaviors or different network topologies by changing the energy function. And the comparison of the energy functions between the different architectures also shows the differences in the behavior at the network level.
> > > I will raise my rating to a certain accept.

---

### Meta-Review · Area_Chair_BJhE · 2022-08-26

**Recommendation:** Accept
**Confidence:** Less certain

**Metareview:**

The submission analyses the (simplified) Transformer architecture from the unfolding optimisation perspective, which was recently used to analyse simpler MLP and CNN models. Four reviewers are positive on the submission results and agree that they potentially can bring new insights and more powerful architectures. AC recommends acceptance.

**Award:**

No

---

### Decision · Program_Chairs · 2022-09-14

Accept